# Large Language Models Suffer From Their Own Output: An Analysis of the Self-Consuming Training Loop

## Abstract

Large Language Models (LLMs) are already widely used to generate content for a variety of online platforms. As we are not able to safely distinguish LLM-generated content from human-produced content, also LLM-generated content is used to train the next generation of LLMs, giving rise to a self-consuming training loop. From the image generation domain, we know that such a self-consuming training loop reduces both quality and diversity of images finally ending in a model collapse. However, it is unclear whether this alarming effect can also be observed for LLMs. Therefore, we present the first study investigating the self-consuming training loop for LLMs. Further, we propose a novel method based on logic expressions that allows us to unambiguously verify the correctness of LLM-generated content, which is difficult for natural language text. Our experimental results for LLMs with up to 49.2M parameters indicate that the self-consuming training loop can produce correct outputs if parameters are chosen correctly, however, the output declines in its diversity depending on the proportion of the used generated data as well as on the diversity of the initial dataset. For our experimental setting, fresh data can slow down this decline, but not stop it. Further, we observe similar results on a real natural language dataset. Given these concerning results, we encourage researchers to study methods to negate this process.

## 1 Introduction

Transformer-based language models have received much attention in the machine learning community in recent years. Especially large language models (LLMs) trained on massive amounts of data from the internet became state of the art in many benchmarks (Brown et al., 2020) and specialized conversational LLM applications like ChatGPT[1] have a massive influence on society already. LLMs can be used for multiple tasks ranging from code generation (Chen et al., 2021; Sobania et al., 2022; Fan et al., 2023) and automated program repair (Sobania et al., 2023) to text summarization (Yang et al., 2023) and teaching assistance (Baidoo-Anu & Ansah, 2023).

Due to their convincing generated outputs, LLMs can be used to generate a large amount of content that is posted online to coding platforms like GitHub and Stackoverflow[2], social media platforms like Reddit[3] and other platforms on the internet. Even academic writing is already being influenced by LLM outputs (Geng & Trotta, 2024; Liang et al., 2024). Such LLM-generated text is often hard to distinguish from human-generated content (Sadasivan et al., 2023) and in turn might unwillingly be used to train the next generation of LLMs. Even paying for human-generated content might not be an option in the future, as workers at paid services like Amazon's Mechanical Turk also use LLMs to produce content (Veselovsky et al., 2023). Consequently, a self-consuming training loop emerges in which future models are trained repeatedly on LLM-generated data from previous generations. This process was first observed for generative models in the image domain (Martínez et al., 2023; Alemohammad et al., 2023; Shumailov et al., 2024; Bertrand et al., 2023). These studies found that this self-consuming training loop leads to a decline in quality and diversity

---

[1]https://openai.com/blog/chatgpt
[2]https://www.microsoft.com/en-us/Investor/events/FY-2023/Morgan-Stanley-TMT-Conference
[3]https://www.vice.com/en/article/jg5qy8/reddit-moderators-brace-for-a-chatgpt-spam-apocalypse

of generated images, ultimately resulting in a so called model collapse. This was also observed for repeatably fine-tuning LLMs leading to a decrease in diversity (Guo et al., 2023; Shumailov et al., 2024). However, it is unclear what happens with LLMs that are trained in such a self-consuming training loop from scratch, as usually done in real-world applications like ChatGPT.

Therefore, we present the first study analyzing the behavior of LLMs trained over many generations in a self-consuming training loop. We conduct experiments on a GPT-style model in different settings and measure both quality and diversity of samples from the trained model at each generation in the self-consuming training loop. The settings differ in the way a dataset is created for each generation (so called data cycles) as well as the proportion of original *real* and LLM-generated *synthetic* data samples. To better analyze the behavior of the trained models we conduct our experiments on a dataset consisting of logic expressions. In contrast to natural language, this logic expressions can be evaluated unambiguously. This allows us to analytically and accurately measure correctness and diversity of the generated samples. Furthermore, to increase the generalizability of our experimental results, we conduct additional experiments on a natural language dataset.

We find that in our controlled experiments with smaller GPT-style models repeatedly training new models with synthetic data from previous models may initially appear to improve quality. However, diversity degenerates and the learned distribution inevitably collapses to a single point. In extreme cases, this happens already after less than 10 generations. Additionally, we find that the speed at which diversity degenerates depends on the data cycle as well as on the proportion of real and synthetic data. The diversity of the initial dataset also plays a role in the rate of decline in diversity. Fresh real data added during the data cycle slows down but can not negate the effects of a self-consuming training loop.

In summary, our main contributions are as follows:

- The first comprehensive empirical study of the self-consuming training loop for LLMs trained from scratch,

- A novel method to unambiguously evaluate quality/correctness and diversity of LLM-generated outputs,

- An in-depth analysis and discussion of the effects and implications of this self-consuming training loop for LLMs.

Following this introduction, Sect. 2 describes the self-consuming training loop and gives an overview of related work. In Sect. 3, we describe our experimental setting. Section 4 presents the experimental results, followed by a discussion in Sect. 5 and limitations in Sect. 6. Section 7 concludes the paper.

## 2 The Self-Consuming Training Loop

Current generations of LLMs are usually trained on large amounts of unstructured data like text and code gathered from the Internet (Brown et al., 2020). Due to their generative nature and capacity those models can in turn be used to generate new *synthetic* data, often indistinguishable from the original *real* data (Sadasivan et al., 2023). This synthetic data ends up back on the internet and thus in the next large dataset, which is used to train the next generation of LLMs. These LLMs in turn produce new content and data, setting in motion a repetitive cycle in which new generations of models are trained each time with a higher proportion of synthetic data from previous generations. We call this process a self-consuming training loop, depicted in Figure 1.

More specifically, consider a dataset $\mathcal{D}_0$ consisting of real data points $x \in X$ sampled from the original distribution $P_X$. A generative model $\mathcal{M}_t$ is trained on this original dataset $\mathcal{D}_0$ until the data is sufficiently fitted, producing the first generation $t = 1$ of generative models. In a self-consuming training loop, a new set of $m$ synthetic data points $\mathcal{S}_t$ is now sampled from the previous generation of generative models $\mathcal{M}_t$. The next generation of generative models $\mathcal{M}_{t+1}$ is then trained from scratch on the new dataset $\mathcal{D}_t$. This self-consuming training loop is repeated until the maximum number of generations $t = T$ is reached. Algorithm 1 in Appendix A presents a pseudo-code of this self-consuming training loop.

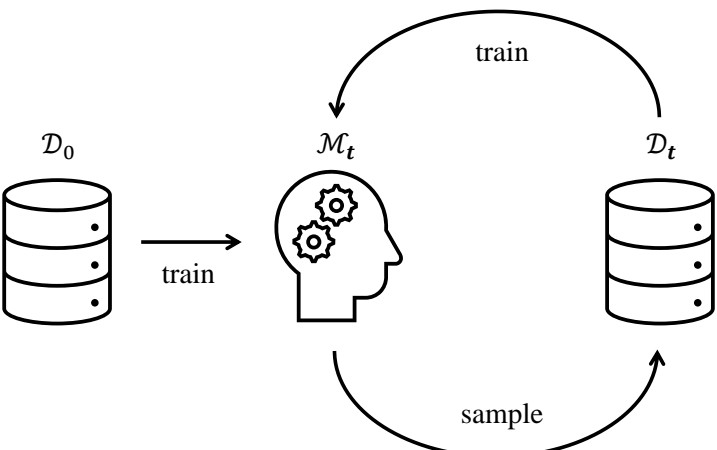

Figure 1: Self-consuming training loop: In the first generation $t = 1$ a model $\mathcal{M}_t$ is initially trained on a real dataset $\mathcal{D}_0$. From this model $\mathcal{M}_t$ a sample $\mathcal{S}_t$ is drawn to build a new dataset $\mathcal{D}_t$. The new dataset $\mathcal{D}_t$ is in turn used to train a new model $\mathcal{M}_{t+1}$ from scratch for the next generation $t + 1$. This process is repeated iteratively until the maximum number of generations $T$ is reached.

This self-consuming training loop differs from self-training to improve LLMs (Huang et al., 2022; Wang et al., 2022; Gulcehre et al., 2023; Singh et al., 2023; Zhang et al., 2024) as the synthetic data is not used intentionally, but rather as a consequence of LLM-generated data published on the internet without being labeled as such.

**Related Work** Current research has analyzed the self-consuming training loop mostly in the context of image generating models. Martínez et al. (2023) first studied the self-consuming training loop for diffusion models and find that the self-consuming training loop leads to a collapse in diversity of the generated images. Other work have given theoretical frameworks for this phenomenon and investigate more complex data cycles in the context of image generation (Alemohammad et al., 2023; Shumailov et al., 2024; Bertrand et al., 2023). Both Alemohammad et al. (2023) and Bertrand et al. (2023) find that fresh real data can lead to stability within the self-consuming training loop.

Shumailov et al. (2024) observe a degeneration of diversity for Variational Autoencoders and Gaussian Mixture Models if some of the output is used again as input. Additionally, the authors perform experiments for iteratively fine-tuning LLMs in a self-consuming way and observe a degradation in quality. They find that degradation is less strong than in the imagine generation context and that keeping some original data helps mitigating this phenomenon. Other contemporary work also analyzed repeated fine-tuning of LLMs in a self-consuming way and observed a decrease in lexical diversity (Guo et al., 2023).

However, new generations of LLMs are usually trained from scratch with web scraped datasets while fine-tuning is mainly done with curated datasets. So the analysis of the self-consuming training loop when training from scratch is of pressing concern. To the best of our knowledge, we are the first to study the behavior of LLMs trained in a self-consuming loop from scratch. Furthermore, self-consuming loops in LLMs have not yet been analyzed with a method that allows to unambiguously evaluate the quality and diversity of generated model output.

## 3  Experimental Setup

We present our logic expression dataset, which is the foundation for the verification of the language model's output. Additionally, we specify the natural language experiments. Furthermore, we explain the data cycles we analyze in our experiments and describe the used model architecture in detail.

### 3.1 Verification with a Logic Expression Dataset

LLMs usually generate natural language texts and their performance is typically measured by using similarity metrics like the BLEU score (Papineni et al., 2002), ROUGE score (Lin, 2004) and BERT score (Zhang et al., 2019) as well as perplexity (Jurafsky & Martin, 2009). However, those metrics rely on measuring similarity of outputs with expected reference data, which can only serve as a proxy for quality of a language model (Callison-Burch et al., 2006; Gehrmann et al., 2023). Consequently, we propose using a dataset consisting of logic expressions. Those expressions can be represented as a sequence making them a good fit for language modeling. In contrast to natural language text, we can easily and systematically evaluate the quality of logic expressions by verifying their correctness. An expression is defined to be syntactically correct, if it can be parsed without an error. The semantic correctness can be evaluated if an expression is either `True` or `False`. If the original dataset only consists of `True` expressions, then a high-quality trained model should also only generate `True` expressions. If the initial dataset consists of equal proportions of both `True` and `False` expressions, a model should generate expressions with the same proportions of `True` or `False` expressions.

We build a logic expression as a tree in a recursive way specified in the function `GenerateLogicExpression`($d$), where $d$ is the desired depth of a logic expression tree. If the desired tree depth is reached when calling the function, a random Boolean is returned (either `True` or `False`). Otherwise, a logic operator (`not,and,or`) is selected and the function is called again recursively with $d - 1$. To build the entire original dataset we use this recursive function to sample new random logic expression trees with a random depth between $d_{min}$ and $d_{max}$ until we have a dataset with $m$ unique expressions that evaluate in a `True` Boolean expression. The complexity of the dataset can be easily controlled by adjusting $d_{max}$. For our experiments we chose a initial dataset size of $m = 10,000$ with expressions of minimum depth $d_{min} = 1$ and maximum depth $d_{max} = 5$. Algorithm 2 in Appendix B describes the generation of our logic expression dataset in pseudo-code.

The resulting logic expression trees can then be saved as strings, e.g., `not ( True and False )`. This allows us to evaluate them in Python using the `eval()` function to test whether they are correct or trigger an error and whether they evaluate to a `True` or `False` Boolean. Additionally, we can encode those strings to a sequence of tokens (`True, False, not, and, or, (, ), <eos>`, where `<eos>` is a stop token and indicates the end of an expression) and use these sequences to train a language model. Examples of logical expressions can be found in Appendix E.1.

### 3.2 Natural Language Experiments

While using the logic expression dataset enables us to design a controlled experimental setting in which we can unambiguously measure both quality and diversity of a trained model, this dataset might not fully cover the characteristics of real textual data. Therefore, we conduct additional experiments using the *tiny Shakespeare dataset* (Karpathy, 2015), a collection of $40,000$ lines of Shakespeare text with over 1.1 million characters. Since there is no way to analytically investigate the correctness of the generated text for this dataset, we focus on the diversity in this experiments. Examples of the original dataset as well as generated samples within the self-consuming training loop can be found in Appendix E.2.

### 3.3 Measuring the Diversity of the Model's Output

A generative model does not only need to generate correct outputs but also a diverse set of outputs. To measure the diversity of a sample from a LLM in the logic expression experiments we use the Levenshtein diversity (Beijering et al., 2008; Wittenberg et al., 2023) in our experiments. This metric calculates the pairwise Levenshtein distance between each expression in the sample normalized by the number of tokens of the longer expression of each pairwise comparison and averaged over the number of pairwise comparisons. If the average pairwise normalized Levenshtein distance is close to zero the diversity within a sample is low. A value closer to one suggests a stronger diversity within a sample. The Levenshtein distance itself is a metric for measuring the distance between two strings and is defined as the number of edits (deletion, addition, substitution) required to change one string to another (Levenshtein et al., 1966). Normally, this

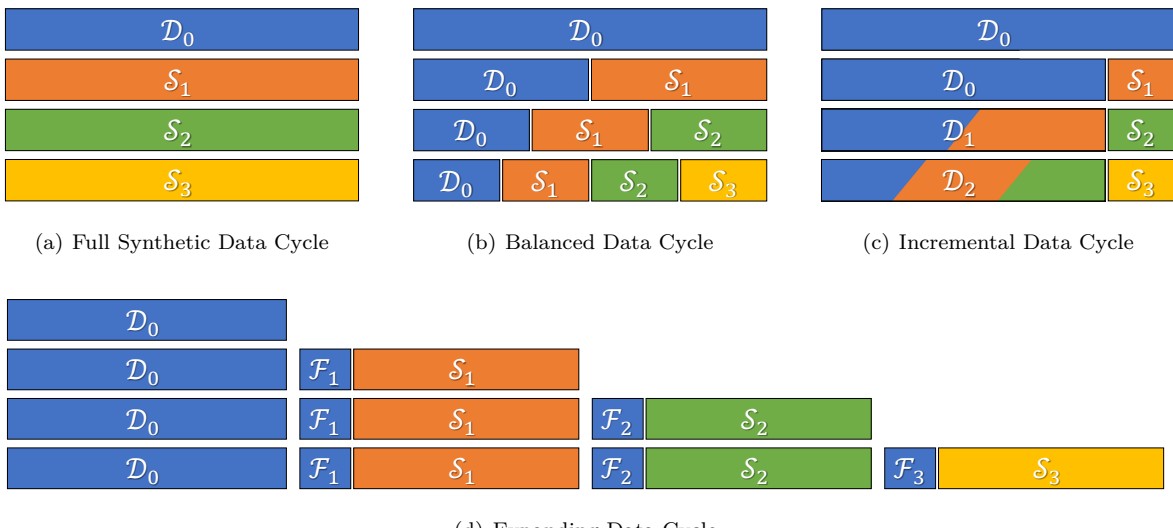

(a) Full Synthetic Data Cycle   (b) Balanced Data Cycle   (c) Incremental Data Cycle

(d) Expanding Data Cycle

Figure 2: Each sub figure represents a different data cycle exemplary for four generations. The first row in each sub figure is the original dataset $\mathcal{D}_0$. The second, third and fourth row depict the dataset $\mathcal{D}_1$, $\mathcal{D}_2$, and $\mathcal{D}_3$ respectively.

edit distance is calculated on a character basis, however, we calculate it on the tokens from our encoding of the logic expressions to accommodate for the different length each token has in character representation.

Additionally, we investigate a wide range of other diversity metrics in Appendix D.8.

To measure the diversity of a sample in the natural language experiments we employ the $n$-gram diversity score (Meister et al., 2023; Li et al., 2023; Padmakumar & He, 2024) which measures the ratio between unique and total $n$-grams in a text. A smaller value indicates less diversity. Furthermore, we investigate embedding based diversity metrics as well as the compression rate (Shaib et al., 2024) and the vocabulary size of the generated samples.

We provide further details for all diversity metrics in Appendix C.

### 3.4 Data Cycles

We define a *data cycle* as the way a dataset $\mathcal{D}_t$ is constructed in generation $t$ from the original data $\mathcal{D}_0$, potential fresh data $\mathcal{F}_t$ from the original distribution, and the generated data from the current and previous generations $\mathcal{S}_{1...t}$. This way of constructing a new dataset may vary and different data cycles are possible. In the image generation domain, previous work suggests that the self-consuming training loop is influenced by the data cycle being used (Alemohammad et al., 2023). Therefore, inspired by their work we conduct our experiments with the following four different data cycles (also depicted in Fig. 2):

**Full Synthetic Data Cycle:** In the most extreme case of a self-consuming training loop a new model $\mathcal{M}_t$ is only trained on the generated data from the last generation so that $\mathcal{D}_{t-1} = \mathcal{S}_{t-1}$. We call this a *full synthetic* data cycle. Normally, new datasets would still contain the original data when they are collected in practice. However, we can use this data cycle to study the most extreme changes in behavior of models from generation to generation.

**Balanced Data Cycle:** We refer to the second data cycle as the *balanced* data cycle. In this data cycle, we construct the new dataset $\mathcal{D}_t$ from equal parts of all the previous samples $\mathcal{S}_{1...t}$ and the original dataset $\mathcal{D}_0$ so that every previous generation contributes $m * \frac{1}{t+1}$ logic expressions to the new dataset $\mathcal{D}_t$ of size $m$.

**Incremental Data Cycle:** The third data cycle we study is called the *incremental* data cycle. In this data cycle, the new dataset is created by a portion $(1 - \lambda)$ of the last dataset $\mathcal{D}_{t-1}$ and a portion $\lambda$ of new

sampled data $\mathcal{S}_t$ so that $\mathcal{D}_t = (1 - \lambda) * \mathcal{D}_{t-1} + \lambda * \mathcal{S}_t$ while holding the size $m$ of the dataset constant each generation. A $\lambda = 1$ would result in a full synthetic data cycle. We chose $\lambda = 0.1$ for our initial experiments and later used different values of $\lambda$ to study the influence of this parameter on diversity.

**Expanding Data Cycle:** The first three data cycles create a dataset of equal size in each generation. In practice, however, the dataset would grow each generation by adding new generated data to the already existing dataset. Additionally, fresh data samples $\mathcal{F}_t$ from the original distribution (e.g. human generated) would also be added at each generation. Consequently, the last data cycle we study is an *expanding* data cycle. At each generation the new dataset is created so that $\mathcal{D}_t = \mathcal{D}_{t-1} + (1 - \lambda) * \mathcal{F}_t + \lambda * \mathcal{S}_t$, where $\lambda$ is the portion of generated data we add each generation and $(1 - \lambda)$ is the portion of fresh real data added at each generation. For this data cycle, we chose $\lambda = 0.9$ for our initial experiments and later used different values of $\lambda$ to study the influence of fresh data in an expanding data cycle on diversity.

### 3.5  Model Architecture and Training

We employ a GPT-style LLM using the open-source implementation *nanoGPT*[4] in our experiments. The model accepts a context of up to 256 tokens and consists of 6 attention layers with 6 attention heads each and an embedding dimensionality of 384, resulting in roughly 10.6 million parameters.

During training we use a batch size of 64 and a dropout rate of 0.2, training for 5,000 iterations minimizing cross entropy loss, starting with a learning rate of $10^{-3}$ decaying to $10^{-4}$, to achieve a sufficient fitting of the training data. We split the dataset in 90% training and 10% validation data. During training we calculate the validation error every 250 iterations and use the model parameters with the lowest validation error as our final model $M_t$ for each generation $t$ in the self-consuming training loop. In our experiments, we do not deduplicate the dataset at each generation $t$. To ensure this has no significant influence on our results, we conduct experiments with deduplication in Appendix D.2. We find that deduplication can reduce the negative effects of a self-consuming training loop but ultimately does not prevent them.

We use the trained model at each generation to sample $10,000$ logic expressions. We also perform experiments for larger amounts of logical expressions ($20,000, 30,000, \& 40,000$) and observe no impact on the results for varying dataset sizes (see Appendix D.5). During sampling, we auto-regressively generate new tokens with temperature 0.8 and feed them back into the model until a stop token `<eos>` is sampled up to a maximum of 200 tokens per expression. We also perform experiments with different temperatures as well as nucleus sampling with different top-p values to get a better understanding of the influence of sampling dynamics in AppendixD.6.

We run the self-consuming training loop for $T = 50$ generations in each experiment.

We use the same model and setup to run the self-consuming training loop for our natural language experiments but tokenize the text on a character level resulting in a vocabulary size of 65. During sampling, we auto-regressively generate text of 1,000 tokens in length until our sample $\mathcal{S}_t$ is of equal size as the original dataset $\mathcal{D}_0$ (approximately 1.1 million tokens per generation). For the expanding data cycle in the natural language experiments we only use a subset of $\mathcal{D}_0$ in order to have enough fresh original data for 50 generations. Consequently, we generate less tokens each generation (approximately $186,000$ tokens per generation).

Experiments are performed on a workstation using consumer NVIDIA graphics cards (TITAN RTX and GeForce RTX 4090) as well as on HPC infrastructure with NVIDIA A100 80GB graphics cards.

## 4  Results

We present our experimental results in terms of correctness and diversity of model outputs trained in a self-consuming training loop.

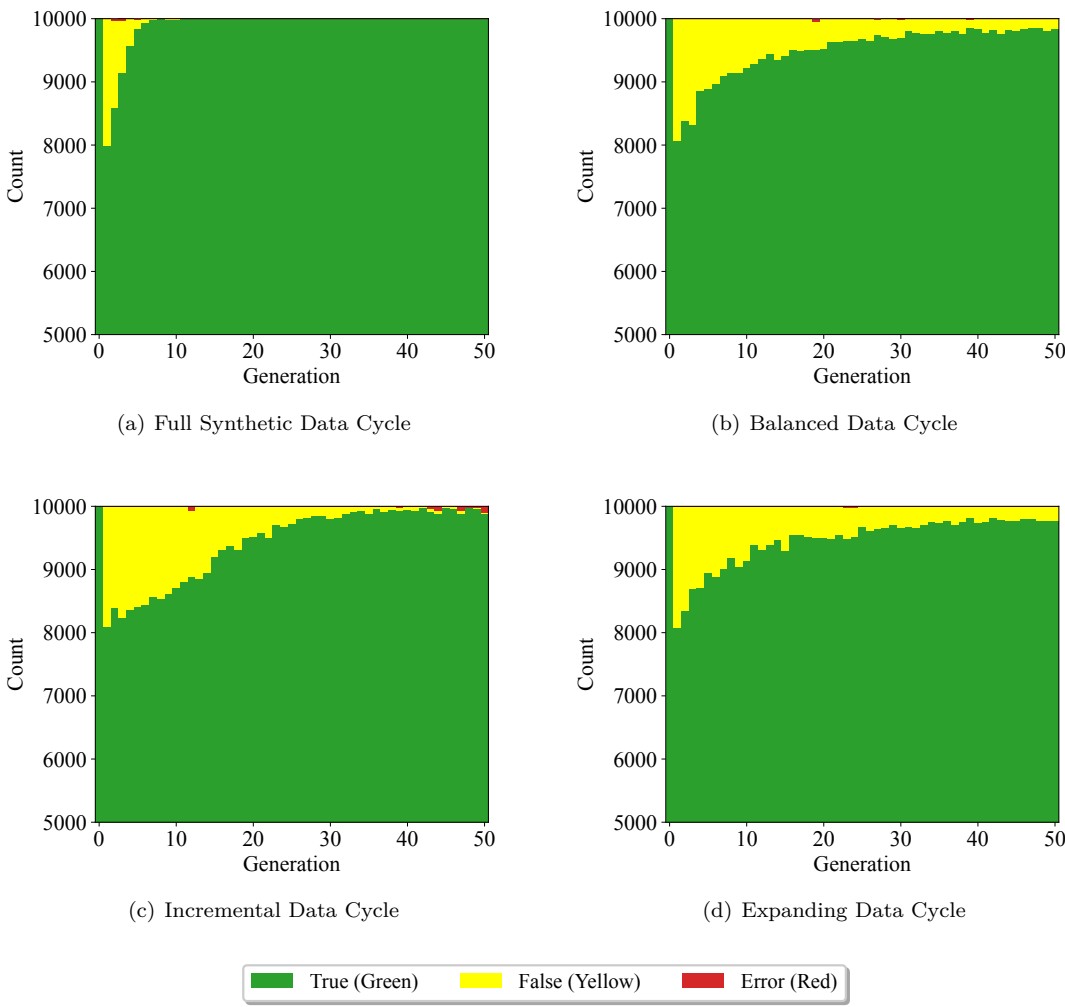

(a) Full Synthetic Data Cycle

(b) Balanced Data Cycle

(c) Incremental Data Cycle

(d) Expanding Data Cycle

True (Green)     False (Yellow)     Error (Red)

Figure 3: The composition of each sample $\mathcal{S}_t$ from model $\mathcal{M}_t$ at generation $t$ (the first bar displays the composition of $\mathcal{D}_0$ consisting of only True expressions) with regards to the number of syntactically and semantically correct expressions. Green indicates syntactically correct expressions that evaluate to True, yellow indicates syntactically correct expressions that evaluate to False, and red indicates syntactically incorrect expressions that result in an error when being parsed. Each subplot displays the results for a different data cycle.

## 4.1   Correctness of Generated Content

We first study the correctness of the expressions within a generated sample $\mathcal{S}_t$ from a model $\mathcal{M}_t$ at each generation $t$. As described in Sect. 3.1, we consider an expression to be syntactically correct if it can be parsed without error. Since the original dataset $\mathcal{D}_0$ only consists of True expressions, we consider a generated expression to be semantically correct if it also evaluates to True. A semantically correct expression is also syntactically correct.

Figure 3 displays the composition of samples $\mathcal{S}_t$ generated during a self-consuming training loop. Each subplot presents the results for a different data cycle: a) full synthetic, b) balanced, c) incremental, and d) expanding. Every bar in a subplot displays the composition of $\mathcal{S}_t$ generated from model $\mathcal{M}_t$ at generation $t$,

---

[4]https://github.com/karpathy/nanoGPT

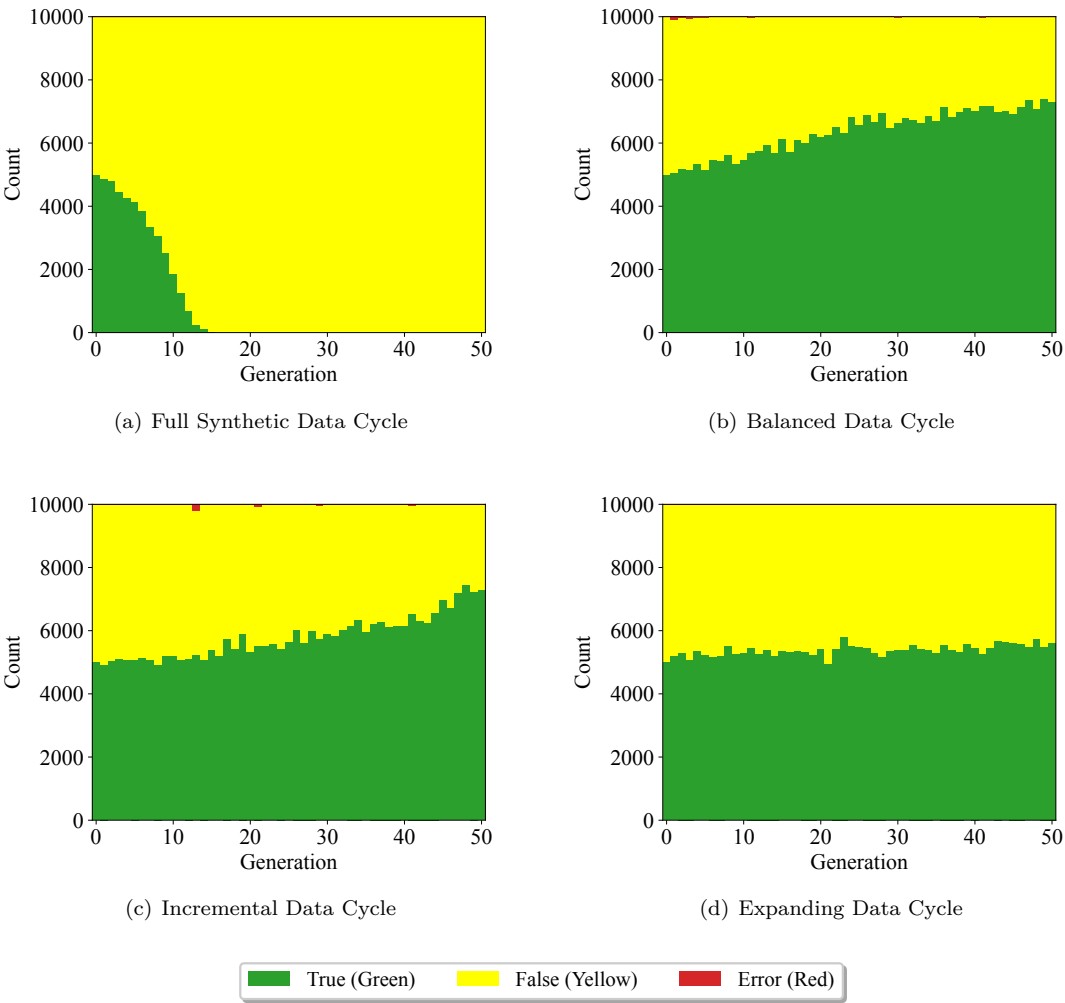

(a) Full Synthetic Data Cycle    (b) Balanced Data Cycle

(c) Incremental Data Cycle    (d) Expanding Data Cycle

True (Green)    False (Yellow)    Error (Red)

Figure 4: The composition of each sample $\mathcal{S}_t$ from model $\mathcal{M}_t$ at generation $t$ (the first bar displays the composition of $\mathcal{D}_0$ consisting of an equal mix of `True` and `False` expressions) with regards to the number of syntactically and semantically correct expressions. Green indicates syntactically correct expressions that evaluate to `True`, yellow indicates syntactically correct expressions that evaluate to `False`, and red indicates syntactically incorrect expressions that result in an error when being parsed. Each subplot displays the results for a different data cycle.

except for the first bar in each subplot which displays $\mathcal{D}_0$. The green portion of a bar indicates the number of syntactically correct expressions that evaluate to `True` in that sample. The yellow part of a bar represents the number of syntactically correct expressions that evaluate to `False`, and the red part of a bar displays the number of syntactically incorrect expressions that result in an error when being parsed.

Overall, we see that only very few expressions are syntactically incorrect. This indicates that the models are sufficiently trained and can correctly learn the syntactic rules of a logic expression. We see a drop of around 20% in the number of semantically correct expressions from the original data $\mathcal{D}_0$ to the first sample $\mathcal{S}_1$ in every data cycle. Interestingly, the number of semantically correct expressions increases afterwards over the course of the self-consuming training loop. The speed at which this number increases depends on the data cycle. We can see the fastest increase for the full synthetic data cycle in which the whole sample consists of `True` expressions by generation $t = 6$. The incremental data cycle takes consistently longer but also nearly reaches this point by generation $t = 36$. While the balanced data cycle has an initially steeper increase in

semantically correct expressions, it does not reach the point of a completely semantically correct sample by the end of 50 generations but comes very close to it. Lastly, the expanding data cycle shows this trend as well.

To better understand the increase in semantically correct expressions, we also study the correctness of the expressions when initialized from an equal mix of `True` and `False` expressions. In this scenario, we would want a model to keep the distribution of those two types of expressions over the generations of a self-consuming training loop.

Similar to before, Figure 4 displays the composition of samples $\mathcal{S}_t$ generated during a self-consuming training loop for different data cycles but initialized from an equal mix of `True` and `False` expressions. Again, we observe only a very small number of syntactically incorrect expressions. However, this time our target distribution is an equal amount of `True` and `False` expressions but all data cycles slowly (very slowly for the expanding data cycle or even rapidly for the full synthetic data cycle) shift to either `True` or `False` expressions.

Overall, this indicates that the distribution of the generated output is shifting during a self-consuming training loop and the rate at which this happens differs by data cycle and the initial dataset. Depending on the task, this may appear to be a positive effect, but it can also have disadvantages like a decline in diversity.

## 4.2 Diversity of Generated Content

Contemporary work in the image generation domain suggests, that the self-consuming training loop comes with a loss of diversity (Martínez et al., 2023; Alemohammad et al., 2023; Bertrand et al., 2023). Therefore, this section studies the diversity within a sample $\mathcal{S}_t$ of model $\mathcal{M}_t$ for each generation $t$.

Figure 5 displays the average pairwise normalized Levenshtein distance over generations of a self-consuming training loop for different data cycles. For each data cycle, we observe a decrease in diversity over the course of generations, with the degree of decrease varying depending on the data cycle. For the full synthetic data cycle, we see a steep decline in diversity with a collapse into a single point by generation $t = 39$. The incremental data cycle is initially stable, but decreases in diversity from the 10th generation onwards, with a decrease of 68% in diversity by the end of the 50 generations. The balanced and expanding data cycle also decrease in diversity, however, this decrease is way slower, with a 30% decrease in diversity for the balanced data cycle and a 22% decrease for the expanding data cycle. While not all data cycles fully collapse in diversity by generation 50, ultimately, we expect all of them to eventually reach zero diversity if the self-consuming training loop is run for enough generations.

We observe a similar decrease in diversity if the initial dataset consists of an equal mix of `True` and `False` expressions. Figure 6 displays the average pairwise normalized Levenshtein distance over generations of the four different data cycles for those experiments. Similar to before, all four data cycles decline in diversity with the full synthetic data cycle decreasing the most and the expanding data cycle decreasing the least. However, the rate of decrease is slightly slower than before. This indicates that the slightly higher diversity of the initial dataset might slow down the effects of self-consuming training loop. Overall, we see the same dynamics in terms of diversity decline as before even if the initial dataset consists of an equal mix of `True` and `False` expressions. Therefore, for the remainder of this section we will focus on experiments initialized with only `True` expressions.

To get further insight into the decline in diversity, we also inspect the number of unique expressions sampled per generation for the full synthetic data cycle. We find that the number of unique expressions is stable at first while the diversity already declines and then rapidly decreases within 5 generations to very few unique expressions (see Appendix D.1).

To better understand the influence of the amount of generated data in each generation on the diversity, we also study the incremental data cycle in more detail with different portions $\lambda$ of new sampled data. Figure 7 displays the average pairwise normalized Levenshtein distance over generations in a self-consuming training loop for the incremental data cycle with different values of $\lambda$. We observe, that the loss in diversity is stronger for a higher share of generated data added in each generation of the self-consuming training loop. Even with as little as $\lambda = 0.25$, we reach the complete loss of diversity within 50 generations. Only for

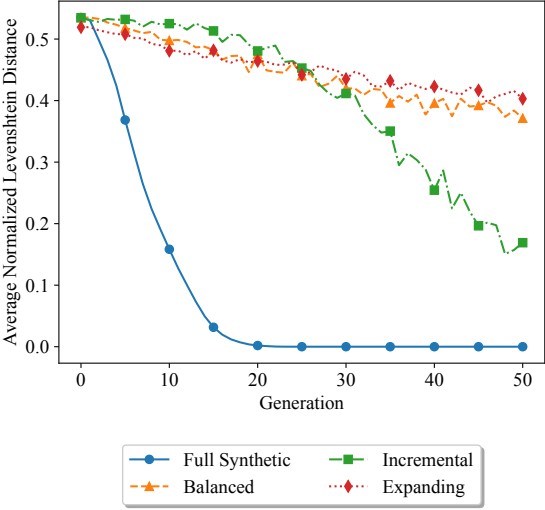

Figure 5: Average pairwise normalized Levenshtein distance over generations of a self-consuming training loop for different data cycles with `True` initialization. Line markers added every 5 generations for better display.

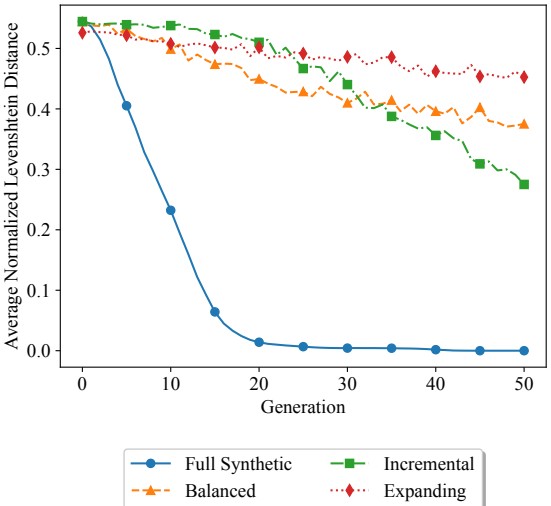

Figure 6: Average pairwise normalized Levenshtein distance over generations of a self-consuming training loop for different data cycles with mixed initialization. Line markers added every 5 generations for better display.

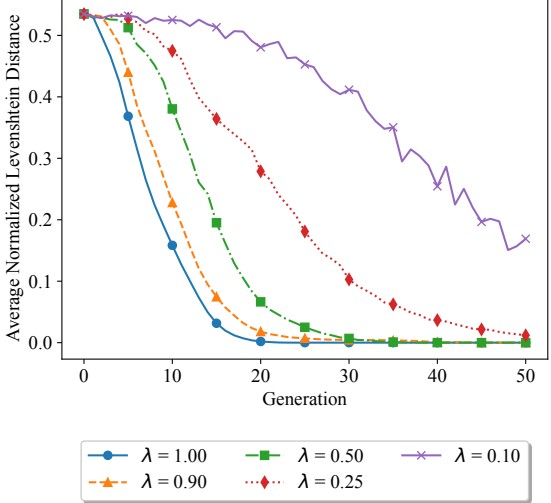

Figure 7: Average pairwise normalized Levenshtein distance over the course of a self-consuming training loop for the incremental data cycle with different portions $\lambda$ of new sampled data. Line markers added every 5 generations for better display.

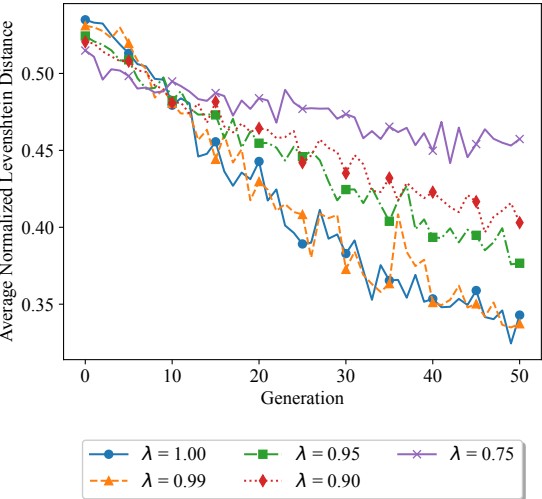

Figure 8: Average pairwise normalized Levenshtein distance over the course of a self-consuming training loop for the expanding data cycle for different portions $\lambda$ of generated data and $(1 - \lambda)$ of fresh data. Line markers added every 5 generations for better display.

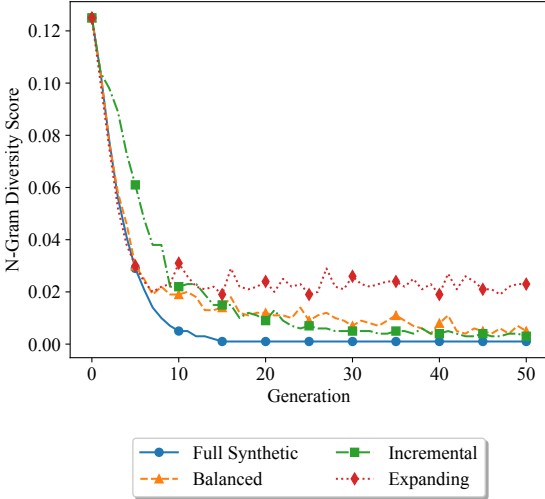 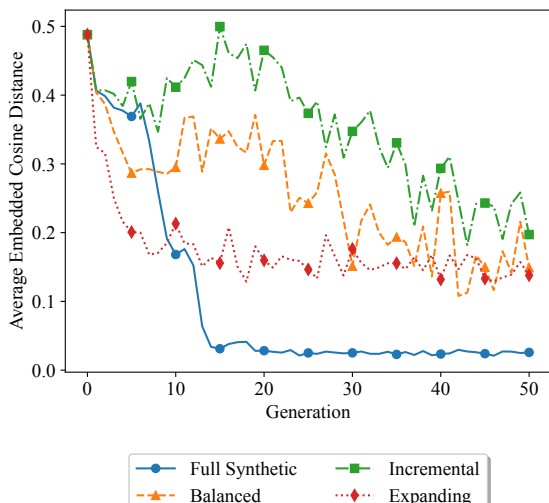

Figure 9: N-gram diversity score over the course of a self-consuming training loop for different data cycles for textual data. The sample size is truncated to equal length for better comparability. Line markers added every 5 generations for better display.

Figure 10: Average cosine distance of embedded text chunks over generations for different data cycles for textual data. The sample size is truncated to equal length for better comparability. Line markers added every 5 generations for better display.

our experimental run with $\lambda = 0.10$ the diversity does not drop to zero, but to a value around 0.17 after 50 generations. However, we see that the speed at which diversity decreases does not scale linear with the amount of generated data. Instead we observe a more exponential increase in speed where larger amounts of generated data lead to an even quicker decay in diversity.

Lastly, we study the expanding data cycle in more detail with regard to the proportion between generated data and fresh data added to the training data at each generation. Figure 8 displays the average pairwise normalized Levenshtein distance over generations in a self-consuming training loop for the expanding data cycle with different portions $\lambda$ of generated data and $(1 - \lambda)$ fresh real data. We observe, that while each configuration declines in diversity, the rate of this decline depends on the proportion between generated and fresh data. The more fresh data is added at each generation, the slower diversity decreases. With no fresh data or only 1% of fresh data the diversity decreases by approximately 36%. If the portion of fresh data is 25% we only observe a decrease in diversity of approximately 11%. This indicates that fresh data cannot stop the self-consuming training loop but enough fresh data can slow down the rate of the decline in diversity.

To provide a more comprehensive view on the diversity decline, we provide additional diversity metrics for the different data cycles including embedding based average pairwise cosine distances, token entropy, average number of abstract syntax tree (AST) depth and AST nodes, and vocabulary distributions in Appendix D.8. Overall, we see the same decline in diversity over all those additional metrics.

### 4.3 Diversity of Generated Natural Language

To extend our results past the domain of logic expression we also study the self-consuming training loop for a real textual dataset. We first investigate the diversity of generated content. Figure 9 displays the word level $n$-gram diversity over generations in a self-consuming training loop for different data cycles. As diversity scores for text are sensitive to text length, we truncated the samples to be of equal size as suggested in (Shaib et al., 2024), to better compare the data cycles with each other. Similar to our results in the logic expression domain, we observe a decline in diversity for all data cycles. As expected, this is most severe for the full synthetic data cycle dropping to zero $n$-gram diversity by generation 15, followed by the incremental

and balanced data cycle. The expanding data cycle retains the highest diversity but also declines strongly over the course of the self-consuming training loop.

However, *n*-gram based metrics might not fully capture the meaning of textual diversity. Thus, we map the generated text to a meaningful embedding vector space which results in a different and more comprehensive view on diversity. Figure 10 displays the average cosine distance between embedded text chunks of $1,000$ token length using a sentences transformer model (Reimers & Gurevych, 2019) over generations of a self-consuming training loop for different data cycles (for details see Appendix C). As before, the samples per data cycle are truncated to equal length. We observe that all data cycles decrease in diversity over generations. The full synthetic data cycle decreases the most and is nearly fully collapsed in diversity by generation 15. Both the balanced and incremental data cycle decrease in diversity with a small increase in diversity between generation 10 and 20 but decreasing from there again with a continued downward trend by generation 50. Interestingly, the expanding data cycle decreases the fastest till around generation 10 and then stabilizes. This is most likely due to the smaller training dataset size (see Sect. 3.5) resulting in a lower starting diversity. Again, this indicates that the diversity of the initial dataset has an effect on the rate of diversity decline.

We also investigate the compression rate and vocabulary size and find similar effects (see Appendix D.9). Overall, our experimental findings for textual data is in line with our findings from the logic expression dataset.

While we cannot evaluate the correctness of the generated text automatically, we manually inspect text samples for a qualitative assessment. We find that while the models produce some hallucinated words in first generations the quality of the generated text is subjectively mostly correct and Shakespeare like. However, the stark decline in diversity quickly leads to repetitive outputs and makes the models unusable, further confirming the effects of the self-consuming training loop. More details and examples of generated text can be found in Appendix E.2.

## 5 Discussion

In our experiments, we found that iteratively re-training LLMs from scratch using self generated data may initially appear to help with correctness of model output. This is sometimes already done in practice where the output of LLMs is used to improve their performance (Huang et al., 2022; Wang et al., 2022; Gulcehre et al., 2023; Singh et al., 2023; Zhang et al., 2024) or generate new training data (Yu et al., 2023). In such a self-training setting, the origin of data is known and researchers can carefully filter and adjust the data. However, the self-consuming training loop is an emerging trend where the data source is not known and LLM-generated content from the internet is unwillingly used to train new models. This trend is concerning as we found experimental evidence in our experiments that this self-consuming training loop might eventually lead to a drastic loss of diversity of a model's output. Our results confirm the findings on the self-consuming training loop in the field of image generation (Martínez et al., 2023; Alemohammad et al., 2023; Shumailov et al., 2024; Bertrand et al., 2023), further highlighting the importance of this issue for generative models and LLMs in particular. These LLMs have an even greater impact on society than image generation models, leading to more LLM-generated data being produced and potentially having a greater impact on society as a whole, as many people already rely on good model performance.

Consequently, as we expect that a self-consuming training loop will harm model performance in the future, researchers and practitioners should carefully choose their data when training their models and test those models sufficiently for diversity. Additionally, the machine learning community at large needs to find ways to deal with this problem as generated text data is already present in new internet scraped datasets. While our results indicate that fresh real data can slow down the self-consuming training loop, other studies suggest that this is not an option as we will run out of real data eventually and have to rely on generated data for future models (Villalobos et al., 2022). Additionally, LLM-generated data has already found its way onto the internet, even infecting academic publications (Geng & Trotta, 2024; Liang et al., 2024), and LLM-generated data may soon overtake real data on the internet.

Furthermore, this LLM-generated data is hard to differentiate from real data (Kreps et al., 2022; Sadasivan et al., 2023; Huschens et al., 2023), making it difficult to curate datasets scraped from the web. This makes

the self-consuming training loop inevitable in the future. Therefore, we advise researchers and practitioners to minimize the effects of the self-consuming training loop whenever possible. This could potentially be done by either collecting diverse data, tracking data sources, deduplication of data, and collecting high quality fresh human data. Additionally, we suggest to further study ways to maintain diversity of generative model outputs like quality diversity methods (Pugh et al., 2015; Fontaine & Nikolaidis, 2021).

## 6 Limitations

A major limitation of our work is the availability of only restricted computational resources. As a result, we ran our experiments only with GPT-style model with a much lower number of parameters than for example GPT-4. Even though we expect that larger models with billions of parameters still suffer from a loss of diversity if trained in a self-consuming training loop, we emphasize that the behavior may vary for those larger models. However, a study for models of those size is not feasible with current compute options nor is it responsible in terms of the expected energy consumption as well as $CO_2$ emissions (Strubell et al., 2019). Nevertheless, we performed additional experiments for the full synthetic data cycle with varying model sizes up to 49.2M parameters in Appendix D.4 and still observe the loss in diversity across smaller and larger models. Therefore, we expect that our experimental findings would also be observable for models of different and also larger size. Nevertheless, energy-responsible future work is still needed to confirm our findings also for models with a few hundred million or even more parameters in the billions.

Second, the main results of this work consist of only one run per experimental configuration. With more runs per experiment the results would be more robust, but conducting more runs for our experiments was simply not feasible with our available computing resources. As all our experiments point towards the same direction, however, we do not believe that this limitation is crucial. Nevertheless, to further mitigate this limitation we performed additional experiments for the full synthetic data cycle with fewer generations over multiple runs and with different initializations in Appendix D.3. These results show that the trend of a decline in diversity still holds true over multiple runs.

Lastly, datasets used to train LLMs cover a wide range of different data sources with different characteristics. Therefore, our results might be slightly different when performed with a real web scraped corpora. However, the experiments with textual data show that our results generalize past logic expressions. Nevertheless, future work should extend our findings to more complex datasets to confirm that the self-consuming training loop behaves similar in those domains.

## 7 Conclusion

This paper studied the behavior of LLMs trained in a self-consuming training loop. In our controlled experiments using a logic expression dataset, we found that iteratively training LLMs from scratch with self generated data can initially appear to help with correctness of model outputs. However, this comes at a price, as the diversity of generated data in our controlled experiments eventually degenerates and –depending on the setting– might collapse into a single point. The rate of this decline in diversity depends on the data cycle creating the training data at each generation and the diversity of the initial dataset. Furthermore, we found that fresh data can slow down this process and preserve diversity of a model output for longer. We found similar results using a natural language dataset, indicating that these concerning results generalize past our controlled experiments. This should encourage researches to carefully monitor their datasets and models to mitigate such a harmful self-consuming training loop in the future.

In future work, we will study how fine-tuning can influence the self-consuming training loop for LLMs and what effects the properties of the initial dataset might have on the decline of diversity. Additionally, we plan to further investigate how the effects of a self-consuming training loop can be slowed down or completely negated.

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

## A    Pseudocode: Self-Consuming Training Loop

Algorithm 1 describes the self-consuming training loop in detail.

---
**Algorithm 1** Self-Consuming Training Loop
---
**Input:** Number of generations $T$, data cycle
**Initialize:** Sample $\mathcal{D}_0$ from $P_X$

**for** $t = 1$ **to** $t = T$ **do**
    Train model $\mathcal{M}_t$ with $\mathcal{D}_{t-1}$
    Sample $\mathcal{S}_t$ from $\mathcal{M}_t$
    Create $\mathcal{D}_t$ from $\mathcal{D}_0$ and $\mathcal{S}_{1\ldots t}$ according to data cycle
**end for**

**Output:** Trained models $\mathcal{M}_{1\ldots T}$, samples $\mathcal{S}_{1\ldots T}$, datasets $\mathcal{D}_{0\ldots T}$

---

## B    Pseudocode: Generation of the Logic Expression Dataset

Algorithm 2 describes the generation of our logic expression dataset in detail.

---
**Algorithm 2** Generate Logic Expression Dataset
---
**Input:** Number of unique expressions $m$, minimum tree depth $d_{min}$, maximum tree depth $d_{max}$
**Initialize:** $\mathcal{D}_0 = \{\}$

**function** GenerateLogicExpression($d$) :
**if** $d = 0$ **then**
    **return** RandomBoolean()
**else**
    $op \leftarrow$ RandomLogicOperator()
    **if** $op =$ 'not' **then**
        **return** $op +$ GenerateLogicExpression($d - 1$)
    **else**
        $leftExpr \leftarrow$ GenerateLogicExpression($d - 1$)
        $rightExpr \leftarrow$ GenerateLogicExpression($d - 1$)
        **return** $leftExpr + op + rightExpr$
    **end if**
**end if**
**end function**

**while** $|\mathcal{D}_0| < m$ **do**
    $i \leftarrow$ RandomIntegerInRange($d_{min}, d_{max}$)
    $expr \leftarrow$ GenerateLogicExpression($d = i$)
    **if** EvaluateBooleanExpression($expr$) $=$ True **then**
        $\mathcal{D}_0 \leftarrow \mathcal{D}_0 \cup \{expr\}$
    **end if**
**end while**

**Output:** $\mathcal{D}_0$

---

## C   Diversity Metrics

We provide more detail on the diversity metrics used for the experiments using logic expression as well as textual data.

The **average normalized Levenshtein distance** (Levenshtein diversity) (Beijering et al., 2008; Wittenberg et al., 2023) used in our logic expression experiments is defined as:

$$\text{Average Normalized Levenshtein Distance} = \frac{1}{|Q|} \sum_{(A,B) \in Q} \frac{d_L(A,B)}{\max(\text{len}(A), \text{len}(B))},$$

where $Q$ is the set of unique pairs of logic expressions within a sample $\mathcal{S}_t$ and $d_L(A,B)$ is the Levenshtein distance between two logic expressions $A$ and $B$.

The **average embedded cosine distance** is defined as:

$$\text{Average Embedded Cosine Distance} = \frac{1}{|Q|} \sum_{(A,B) \in Q} 1 - \text{sim}_{\cos}(f(A,\theta), f(B,\theta)),$$

where $Q$ is the set of unique pairs of logic expressions within a sample $\mathcal{S}_t$ and $\text{sim}_{\cos}(f(A,\theta), f(B,\theta))$ is the cosine similarity between the embeddings of two logic expressions $A$ and $B$ using a sentence-transformer model $f(x,\theta)$ (Reimers & Gurevych, 2019). We use the `all-MiniLM-L6-v2` from Hugging Face.[5] For the experiments with textual data, $A$ and $B$ are text chunks of $1,000$ tokens as generated during sampling.

The **token-level entropy** (Shannon, 1948) is defined as:

$$H = - \sum_{i=1}^{M} p_i \log_2 p_i,$$

where $M$ is the number of unique tokens in $\mathcal{S}_t$ and $p_i$ is the empirical probability of the $i$-th token.

The **n-gram diversity score** (Meister et al., 2023; Li et al., 2023; Padmakumar & He, 2024) used to investigate the diversity of textual data is defined as the number of unique $n$-grams divided by the total number of $n$-grams in a sample $\mathcal{S}_t$:

$$\text{N-Gram Diversity Score} = \sum_{n=1}^{4} \frac{\# \text{ unique } n\text{-grams in } \mathcal{S}_t}{\# \ n\text{-grams in } \mathcal{S}_t}$$

The **compression rate** as a measure for diversity of a text (Shaib et al., 2024) is defined as the size of a sample $\mathcal{S}_t$ divided by the size of the same sample compressed by a compression algorithm (e.g. gZip):

$$\text{Compression Rate} = \frac{\text{size of } \mathcal{S}_t}{\text{compressed size of } \mathcal{S}_t}$$

## D   Additional Experimental Results

### D.1   Unique Expressions Generated

To better understand the decline of diversity, we inspect the number of unique expressions sampled at each generation. We focus on the full synthetic data cycle, as this data cycle fully collapses to a single point within 50 generations.

---

[5]https://huggingface.co/sentence-transformers/all-MiniLM-L6-v2

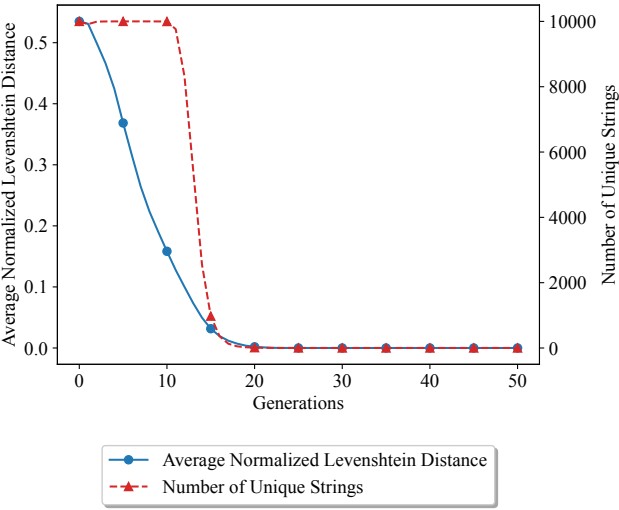 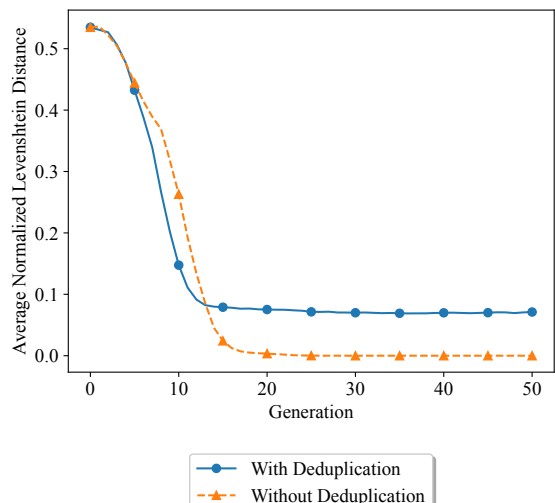

Figure 11: Average pairwise normalized Levenshtein distance (left y-axis) and number of unique expressions in a sample (right y-axis) over generations of a self-consuming training loop for the full synthetic data cycle. Line markers added every 5 generations for better display.

Figure 12: Average pairwise normalized Levenshtein distance over generations of a self-consuming training loop for the full synthetic data cycle with and without deduplication. Line markers added every 5 generations for better display.

Figure 11 displays the number of unique expressions (right y-axis) generated at each generation of a self-consuming training loop in comparison to the diversity measured as the average pairwise normalized Levenshtein distance (left y-axis). We observe that in the beginning the number of unique expressions stays stable while the diversity is already in a steep decline. Only by generation $t = 10$ the number of unique individuals starts to decline rapidly and by generation $t = 15$ only around 1000 unique expressions are sampled while the decline in diversity slows down. In generation $t = 20$ the number of unique individuals is below 10 and in generation $t = 26$ only a single individual is sampled at each generation. Consequently, the diversity is collapsed to zero.

This shows that it is not enough to only track unique outputs of a model, but that diversity needs to be tracked as well. By the time a decrease in unique results is noticeable, the diversity already decreased significantly.

### D.2 Impact of Deduplication during Training

As observed in Appendix D.1, the number of unique expressions generated declines over the course a self-consuming training loop. While the decline in diversity is already observed before the number of unique expressions starts to drop, this might further increase the problem at hand. Conversely, deduplication of training data might have a positive effect on the self-consuming training loop. Therefore, we conduct additional experiments for the logic expression dataset and the full synthetic data cycle using deduplication of the training data.

Figure 12 displays the diversity measured as the average pairwise normalized Levenshtein distance over generations for a self-consuming training loop with and without deduplication of training data. We observe that both runs behave similar until the number of unique expressions start to decline rapidly (see Fig. 11). At this point the deduplication somewhat stabilizes the diversity. However, by that point the diversity of the output of the generated model is already very low.

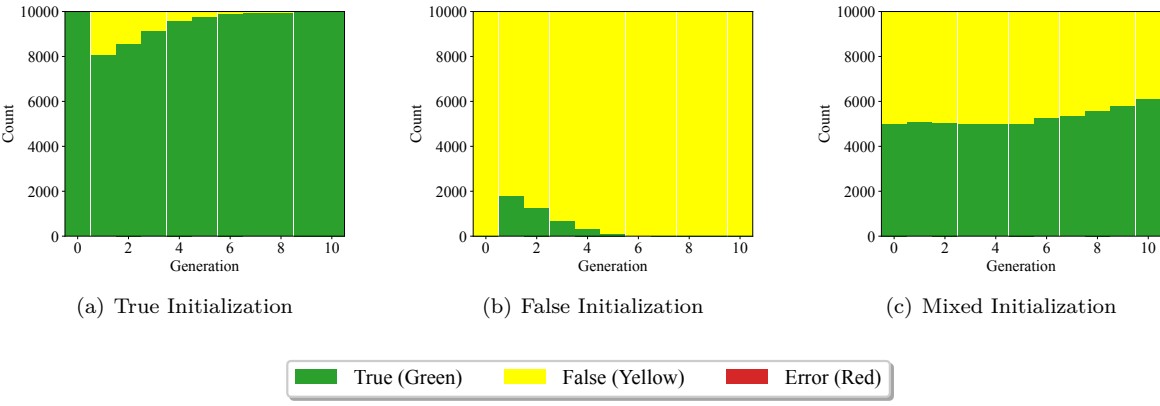

(a) True Initialization       (b) False Initialization       (c) Mixed Initialization

Figure 13: The composition of each sample $\mathcal{S}_t$ from model $\mathcal{M}_t$ at generation $t$ (the first bar displays the composition of $\mathcal{D}_0$) with regards to the number of syntactically and semantically correct expressions averaged over 5 runs. Green indicates syntactically correct expressions that evaluate to `True`, yellow indicates syntactically correct expressions that evaluate to `False`, and red indicates syntactically incorrect expressions that result in an error when being parsed. Each subplot displays the results for the full synthetic data cycle with different initializations (`True`, `False`, Mixed).

Thus, deduplication can indeed help with the negative effects of a self-consuming training loop, however, in our experiments we only find a low influence.

## D.3 Impact of Initialization

So far we only investigated the self-consuming training loop in different scenarios for a single run. To check whether our findings also hold for different initializations of data across multiple runs, we conduct further experiments. Specifically, we analyze the full synthetic data cycle with the original dataset $\mathcal{D}_0$ only consisting of (1) `True` expressions, (2) `False` expressions, and (3) an equal mixture of both. The results from Sect. 4 indicate that the effect of the self-consuming training loop for the full synthetic data cycle are already clearly visible in the first 10 generations. Therefore, we conduct experiments with only 10 generations. With this saved computing time, we conduct 5 independent runs for each initialization.

Figure 13 displays the composition of samples $\mathcal{S}_t$ generated during a self-consuming training loop. Each subplot presents the composition for a different initialization for the full synthetic data cycle averaged over 5 independent runs: a) only `True` expressions, b) only `False` expressions, and c) half `True` expressions and half `False` expressions. Every bar in a subplot displays the composition of $\mathcal{S}_t$ generated from model $\mathcal{M}_t$ at generation $t$, except for the first bar in each subplot which displays $\mathcal{D}_0$. The green portion of a bar indicates the number of syntactically correct expressions that evaluate to `True` in that sample. The yellow part of a bar represents the number of syntactically correct expressions that evaluate to `False`, and the red part of a bar displays the number of syntactically incorrect expressions that result in an error when being parsed.

Similar to our results in Sect. 4.1, we observe an initial drop of `True` expressions by 20% in favor of `False` expressions for the initialization with only `True` expressions in the first generation. Afterwards, the number of `True` expressions in a sample increases again over the course of generations. For the initialization with only `False` expressions, we observe a similar effect, only in the opposite direction. The number of `False` expression first drops by around 20% and then increases again until the whole sample consists of only `False` expressions. When initialized with an equal amount of `True` and `False` expressions, we can observe that the proportion of both expression types first stays stable and then by generation $t = 6$ starts to slightly shift towards `True` expressions. Upon further inspection of the 5 runs, we find that one run quickly shifts towards only `True` expressions, one run shifts quickly to only `False` expression, one run shifts slowly to `True`

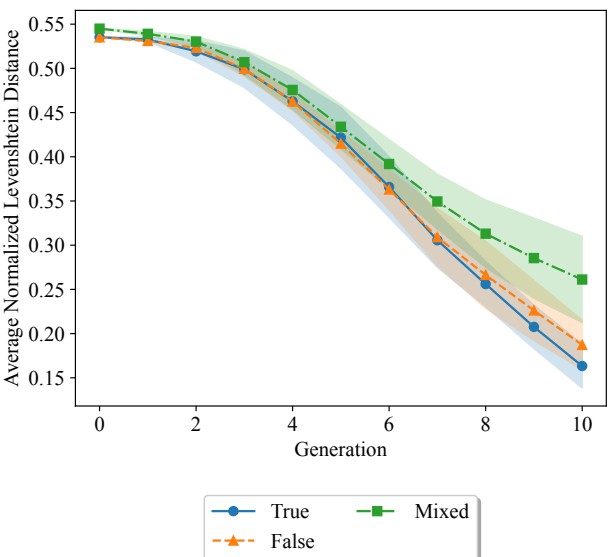

Figure 14: Average pairwise normalized Levenshtein distance over generations of a self-consuming training loop for the full synthetic data cycle with different initializations. Displayed is the mean and standard deviation at each generation for each initialization over 5 runs. Line markers added every 5 generations for better display.

expressions, and two runs maintain equal proportions between `True` and `False` expressions across the 10 generations.

Figure 14 displays the diversity measured in the average pairwise normalized Levenshtein distance for the three different initializations. Displayed is the mean and standard deviation of five runs over the number of generations. We observe that the diversity declines for all three types of initialization, further confirming our results from Sect. 4.2. The initializations with only `True` and only `False` expressions decline at around the same speed. The initialization with equal proportions of `True` and `False` expressions declines a bit slower but shows the same effect.

Overall these findings indicate that the effects of the self-consuming training loop apply to different initializations across multiple runs, partially mitigating the limitations mentioned in Sect. 6.

### D.4    Impact of Model Size

To investigate if our results can be generalized to different model sizes and architectures we repeated our experiments for the full synthetic data cycle for 10 generations with multiple different model sizes. We conducted experiments for models with parameters ranging from 3.2 million to 49.2 million. Table 1 describes the details of those models.

Figure 15 displays the diversity measured in the average pairwise normalized Levenshtein distance for different model sizes. Similar to our results in Sect. 4.2 we observe a steady decline in diversity over generations regardless of model size. While the decrease in diversity happens quicker for the smaller model it is also clearly present in the larger models. Therefore, we argue that the effects of a self-consuming training loop is also present for different model sizes.

### D.5    Impact of Dataset Size

While we used $10,0000$ logic expressions for our main experiments, we conducted additional experiments for larger amounts of logical expressions ($20,000$, $30,000$, and $40,000$) to analyze the impact of varying amounts

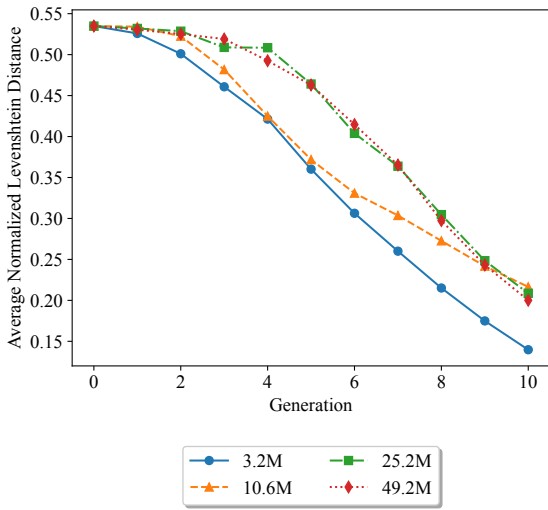 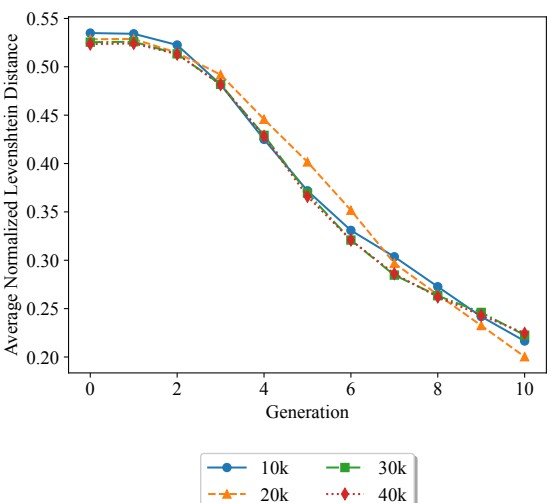

Figure 15: Average pairwise normalized Levenshtein distance over generations of a self-consuming training loop for the full synthetic data cycle with different model sizes. Line markers added every 5 generations for better display.

Figure 16: Average pairwise normalized Levenshtein distance over generations of a self-consuming training loop for the full synthetic data cycle with different dataset sizes. Line markers added every 5 generations for better display.

Table 1: Details of the model architectures for the additional experiments. Displayed is the total number of parameters in millions, the number of layers, number of attention heads per layer, and the embedding dimensionality.

| #PARAMETERS | #LAYERS | #ATTENTION HEADS | EMBEDDING DIMENSION |
|---|---|---|---|
| 3.2M | 4 | 4 | 256 |
| 10.6M | 6 | 6 | 384 |
| 25.2M | 8 | 8 | 512 |
| 49.2M | 10 | 10 | 640 |

of training data on the self-consuming training loop. Specifically, we conducted experiments for the full synthetic data cycle over 10 generations.

Figure 16 displays the diversity measured in the average pairwise normalized Levenshtein distance for different dataset sizes. We observe that for all dataset sizes the diversity decreases over the course of generations. This indicates that the effect of the self-consuming training loop occurs regardless of dataset size.

## D.6 Impact of Sampling Dynamics

Data sampled from a generative model can differ depending on the temperature used during sampling. A lower temperature leads to a more deterministic output while a higher temperature results in a more random output. A more deterministic output might lead to lower diversity while a more random output might promote diversity. However, a more random output might also produce more errors leading to different undesirable effects in a self-consuming training loop. Therefore, we investigate the self-consuming training loop in a full synthetic data cycle for different temperature values using the logic expression dataset initialized with only `True` expressions.

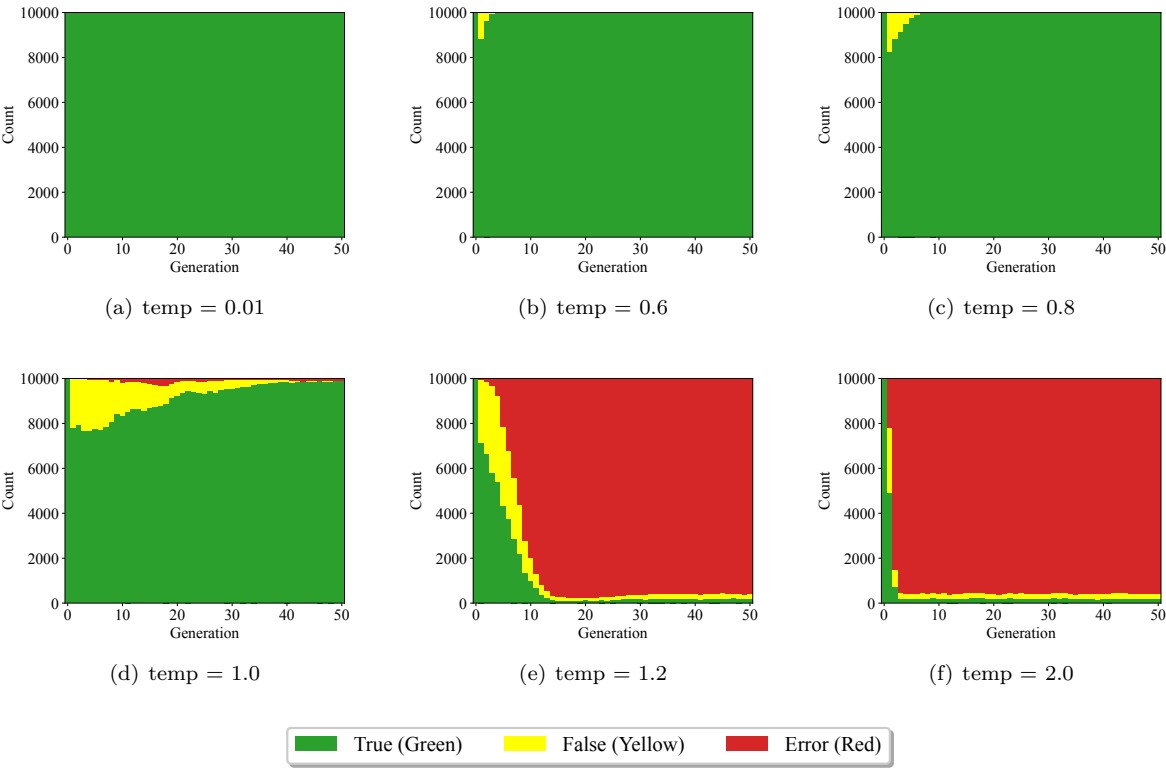

Figure 17: The composition of each sample $\mathcal{S}_t$ from model $\mathcal{M}_t$ at generation $t$ (the first bar displays the composition of $\mathcal{D}_0$) with regards to the number of syntactically and semantically correct expressions. Green indicates syntactically correct expressions that evaluate to `True`, yellow indicates syntactically correct expressions that evaluate to `False`, and red indicates syntactically incorrect expressions that result in an error when being parsed. Each subplot displays the results for a full synthetic data cycle using different temperature values for sampling.

Figure 17 displays the composition of samples $\mathcal{S}_t$ generated during a self-consuming training loop for the full synthetic data cycle. Each subplot presents a run with a different temperature value. When comparing to the original temperature of 0.8 we observe that lower temperature values lead to a quicker shift of only `True` expressions (in the extreme case of temp=0.01 within the first generation). Conversely, a larger temperature of 1.0 leads to a slower, yet noticeable, shift in distribution but with more syntactically incorrect expressions. However, for even larger temperatures we observe a different effect: Instead of producing more `True` expressions we observe an increase in syntactically incorrect expressions until most of the sampled data consists of those. In the extreme case of temp=2.0 this happens within three generations.

This behavior can also be observed when looking at the diversity of the sampled expressions. Figure 18 displays the diversity measured as average pairwise normalized Levenshtein distance for the full synthetic data cycle with different temperature values. While lower temperature values lead to a faster decline in diversity, larger values lead to the opposite. A temperature value of 1.0 slows down the decline in diversity and the larger temperature values 1.2 and 2.0 actually promote diversity over generations. However, this comes at the price of correctness (see Figure 17) as errors accumulate over the course of a self-consuming training loop.

So far we considered the entire probability distribution over all tokens for sampling due to the small vocabulary size of the logic expression dataset. However, in practice only a portion of the tokens is considered during each sampling step, for example only the most likely tokens with a combined probability of top-p, called nucleus sampling (Holtzman et al., 2020). Therefore, we investigate the effects of nucleus sampling

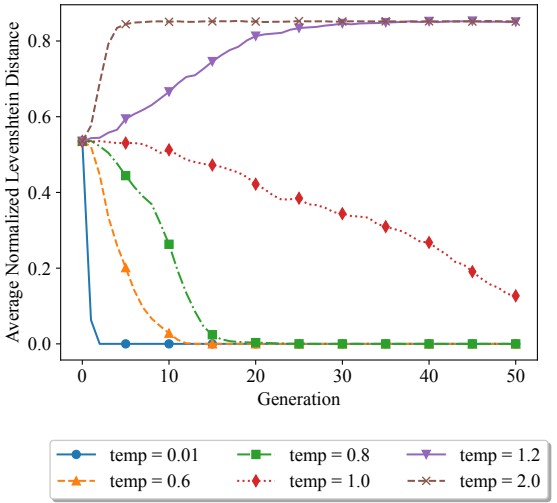 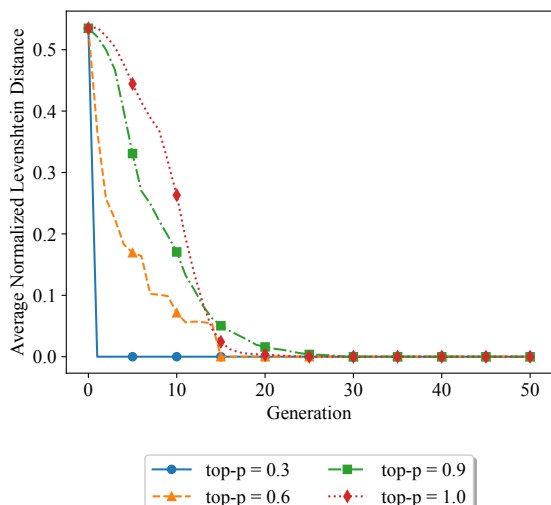

Figure 18: Average pairwise normalized Levenshtein distance over generations of a self-consuming training loop for the full synthetic data cycle with different temperature values for sampling. Line markers added every 5 generations for better display.

Figure 19: Average pairwise normalized Levenshtein distance over generations of a self-consuming training loop for the full synthetic data cycle with different top-p values for sampling. Line markers added every 5 generations for better display.

with different top-p values on the self-consuming training loop. As this narrows the distribution even more when sampling, we expect that small values of top-p lead to a stronger diversity loss.

Figure 19 displays diversity measured in average pairwise normalized Levenshtein distance for the full synthetic data cycle with different values of top-p and a temperature value of 0.8. We observe that smaller values of top-p indeed increase the speed of decline in diversity. While the difference between top-p=1.0 and top-p=0.9 is marginal, top-p=0.6 leads to a notably faster decline and for top-p=0.03 the diversity is collapsed immediately. Therefore, we find that lower values of top-p lead to a faster decline while the extreme degree of this effect is most likely due to the nature of the logic expression dataset.

In general, we find that the exact way of sampling has an effect on the self-consuming training loop, by either increasing the speed of the decline in diversity or by adding errors that accumulate over time. Both outcomes are not desirable for a generative model and should therefore be carefully considered.

### D.7 Self-Consuming Training Loop of Multiple Models in Parallel

Our experiments focus on single models in a self-consuming training loop. However, in the LLM ecosystem there exists a variety of different models that all generate data in parallel. This results in potential synthetic training data accumulated from all those different models. While it is not tractable to realistically simulate this in a controlled experiment, we perform a small scale version of this setting. Starting from the same initial logic expression dataset $\mathcal{D}_0$, we train three models of different size (3.2M, 10.6M, and 25.2M, see Appendix D.4) at each generation in the full synthetic data cycle and sample one third of the new training dataset from each model such that $\mathcal{D}_t = \mathcal{S}_{t_{3.2M}} + \mathcal{S}_{t_{10.6M}} + \mathcal{S}_{t_{25.2M}}$.

Figure 20 displays the diversity measured as average pairwise normalized Levenshtein distance of the samples of the three models over generations. We observe that all three models simultaneously decline in diversity with very similar behavior to each other. The rate of the decrease in diversity is slightly slower than it is for each model in isolation (compare Fig. 15). This indicates that the self-consuming training loop could be slowed down by the additional diversity added from multiple models, but it is still present.

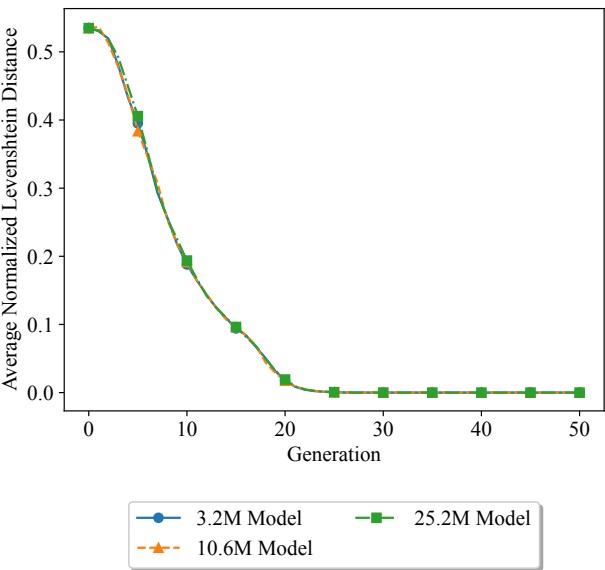

Figure 20: Average pairwise normalized Levenshtein distance over generations of a self-consuming training loop for the full synthetic data cycle with three different models participating in the same loop. Line markers added every 5 generations for better display.

### D.8    Additional Metrics for the Logic Expression Dataset

In order to get a more comprehensive view on the decline of diversity in a self-consuming training loop, we provide results using the additional metrics embedding-based average cosine distance, token entropy, AST statistics, and vocabulary distributions for all data cycles with `True` and mixed initialization.

Figure 21 and 22 display the average pairwise cosine distance of logic expressions embedded into vector space using a sentence transformer model (see Appendix C) for all data cycles over generations of a self-consuming training loop with `True` and mixed initialization respectively. Similar to our findings in Sect. 4.2, we find that the full synthetic data cycle decreases in diversity the fastest, followed by the balanced and incremental data cycle. Due to the different initial datasets, the diversity for the expanding data cycle starts at a slightly lower level but ends on a higher diversity at generation 50 compared to the other data cycles.

Figure 23 and 24 display the token entropy per data sample $\mathcal{S}_{\sqcup}$ over the generations of a self-consuming training loop for all data cycles and `True` and mixed initializations respectively. Again, we observe a decline for all data cycles with the rate of decline depending on the data cycle. Additionally, we observe that the decline in token entropy is larger for the `True` initialization compared to the mixed initialization for all data cycles.

Additionally, we investigate the structure of the generated logic expressions by presenting the average AST depth and number of AST nodes of logic expressions sampled per generation. Figures 25 and 26 display the AST depth for all data cycles using `True` and mixed initialization, respectively. We observe that for all data cycles and initializations the average tree depth increases. The full synthetic data cycle converges before generation 10 in line with the collapse in diversity observed in the other metrics. The average number of AST nodes, displayed in Figs. 27 and 28, follows a similar pattern increasing over the number of generations and converging for the full synthetic data cycle around generation 15. Again, this supports the observed decline in diversity on other metrics.

Lastly, we investigate the distribution of tokens in our vocabulary over generations. Figures 29 and 30 display the mean relative frequency a token appears in a sampled expression. The shaded regions describe the standard deviation for each token. We observe a shift in distribution for all data cycles and both

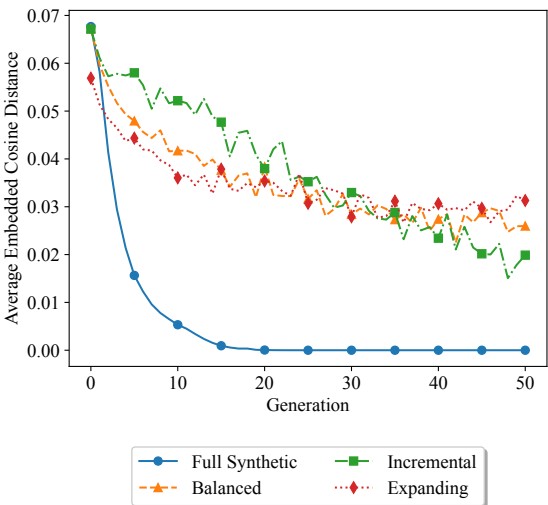

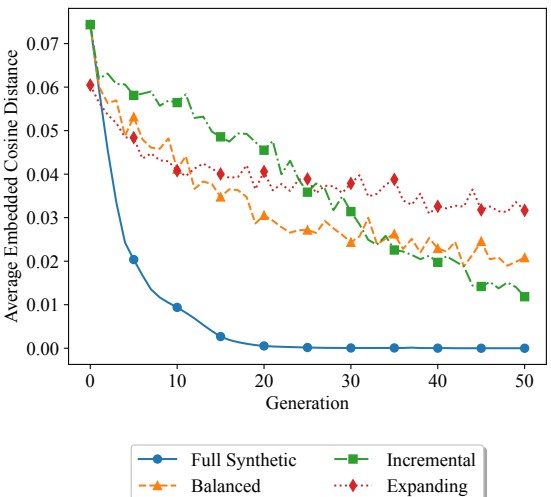

Figure 21: Average cosine distance of embedded logic expressions over the course of a self-consuming training loop for different data cycles with `True` initialization. Line markers added every 5 generations for better display.

Figure 22: Average cosine distance of embedded logic expressions over the course of a self-consuming training loop for different data cycles with mixed initialization. Line markers added every 5 generations for better display.

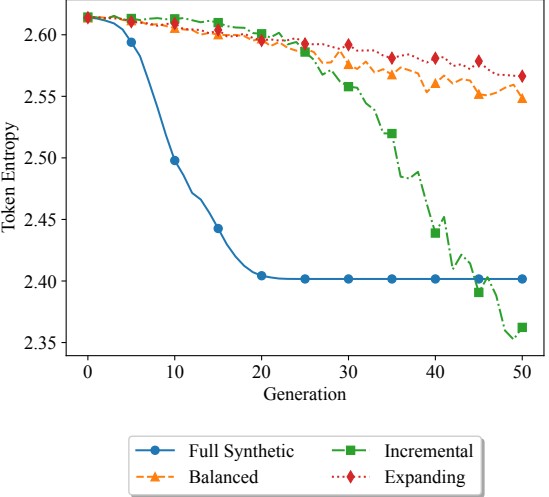

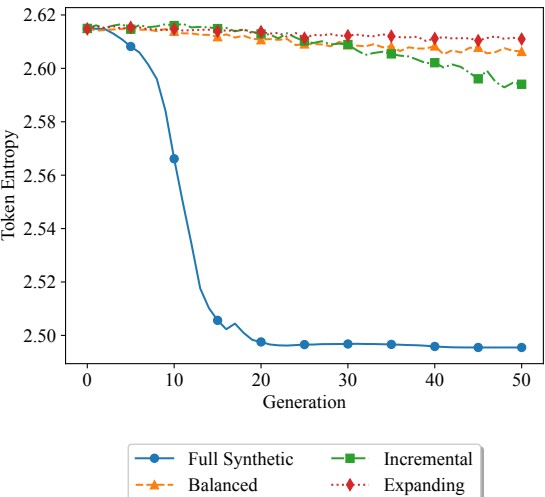

Figure 23: Token entropy of sampled expressions over the course of a self-consuming training loop for different data cycles with `True` initialization. Line markers added every 5 generations for better display.

Figure 24: Token entropy of sampled expressions over the course of a self-consuming training loop for different data cycles with mixed initialization. Line markers added every 5 generations for better display.

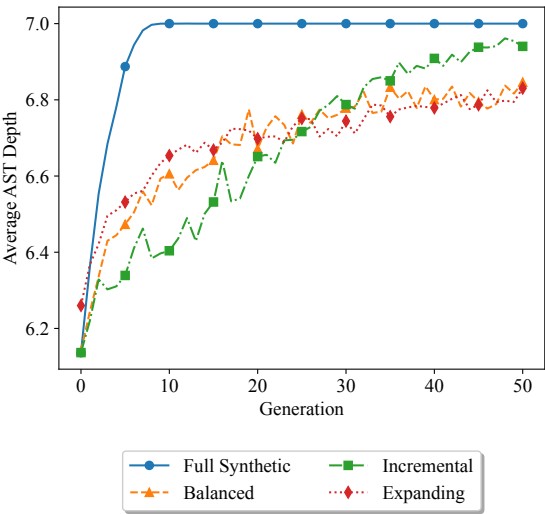

Figure 25: Average AST depth of logic expressions over the course of a self-consuming training loop for different data cycles with `True` initialization. Line markers added every 5 generations for better display.

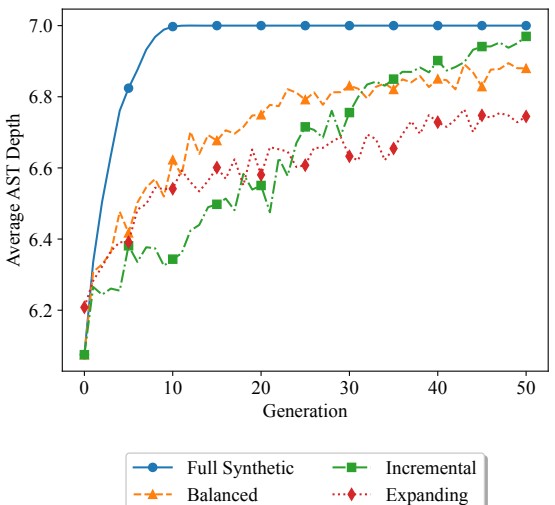

Figure 26: Average AST depth of logic expressions over the course of a self-consuming training loop for different data cycles with mixed initialization. Line markers added every 5 generations for better display.

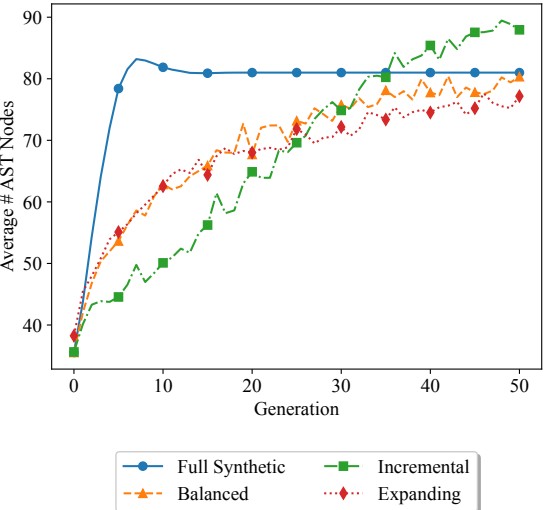

Figure 27: Average number of AST nodes of logic expressions over the course of a self-consuming training loop for different data cycles with `True` initialization. Line markers added every 5 generations for better display.

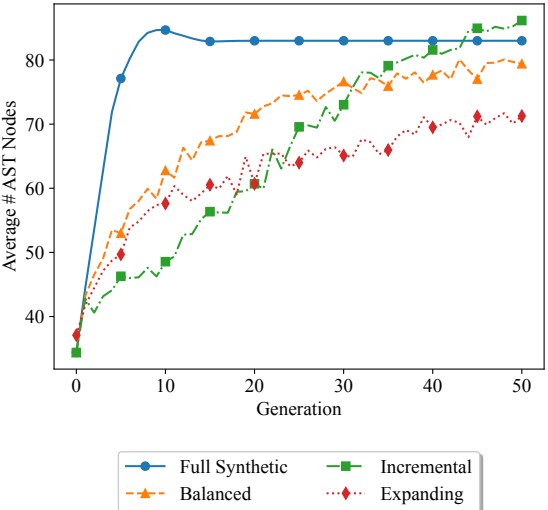

Figure 28: Average number of AST nodes of logic expressions over the course of a self-consuming training loop for different data cycles with mixed initialization. Line markers added every 5 generations for better display.

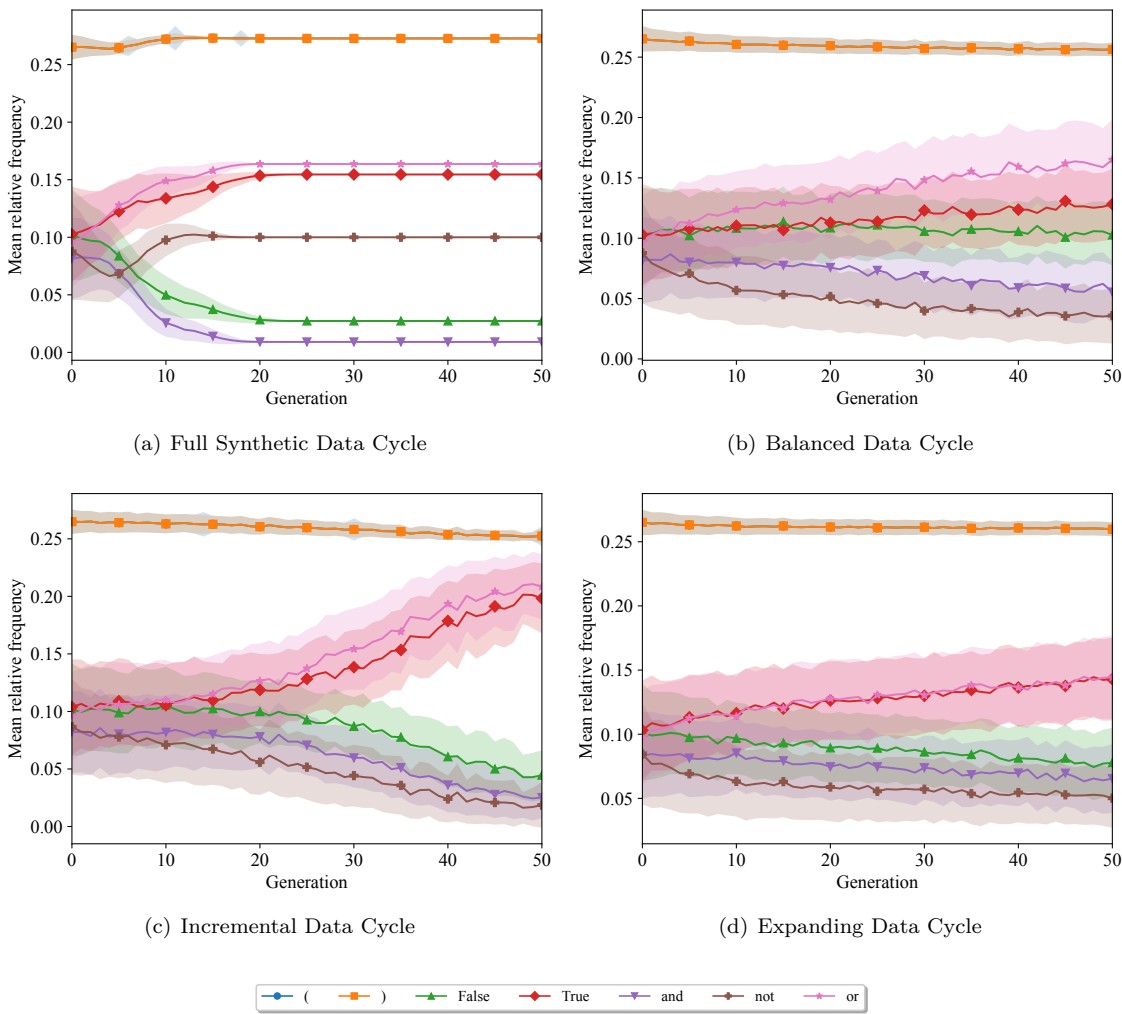

(a) Full Synthetic Data Cycle

(b) Balanced Data Cycle

(c) Incremental Data Cycle

(d) Expanding Data Cycle

Figure 29: The distribution of vocabulary tokens over the course of a self-consuming training loop for different data cycles with `True` initialization. Displayed is the mean frequency of occurrence per token in a sampled logic expression over generations. The shaded regions indicate the standard deviation for each token. Line markers added every 5 generations for better display.

initializations. We also observe that the standard deviations are shrinking over the course of the self-consuming training loop indicating a loss of diversity as observed with other metrics. This is particularly evident for the full synthetic data cycle. Interestingly, the shift is less profound for the mixed initialization. Again, this supports the hypothesis that the diversity of the initial dataset has an impact on the rate of decline. The shift in distribution and decrease in diversity is lowest for the expanding data cycle with mixed initialization confirming the findings from above.

Overall, across all additional metrics we observe a shift in distribution and a decline in diversity depending on the data cycle and initialization further confirming our findings.

## D.9 Additional Diversity Metrics for the Natural Language Dataset

In addition to our results in section 4.3 we also inspect the compression rate (Shaib et al., 2024) and the vocabulary size of samples obtained in the self-consuming training loop for textual data. Once again we truncate the data samples to be of equal size for better comparability between the different data cycles.

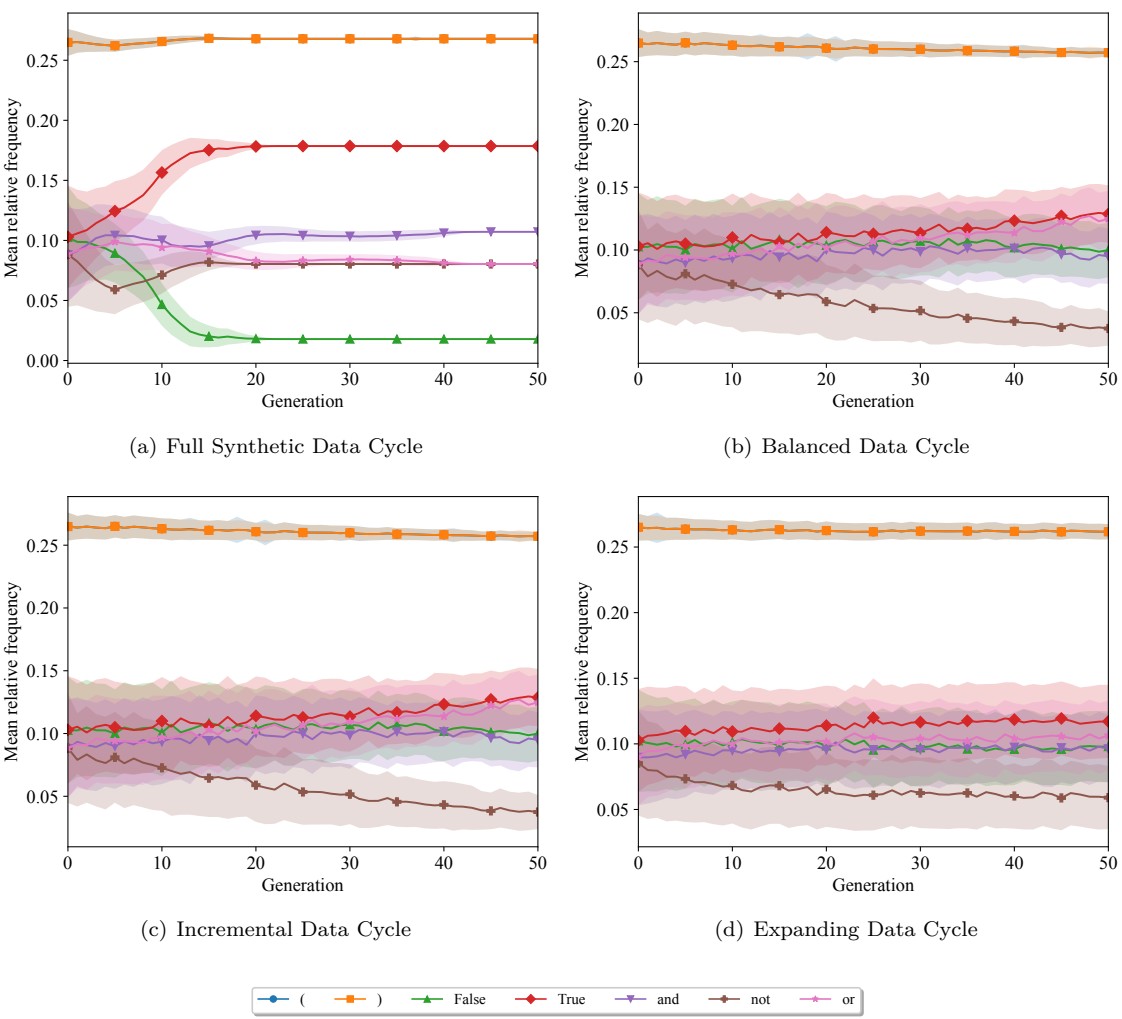

(a) Full Synthetic Data Cycle

(b) Balanced Data Cycle

(c) Incremental Data Cycle

(d) Expanding Data Cycle

Figure 30: The distribution of vocabulary tokens over the course of a self-consuming training loop for different data cycles with mixed initialization. Displayed is the mean frequency of occurrence per token in a sampled logic expression over generations. The shaded regions indicate the standard deviation for each token. Line markers added every 5 generations for better display.

Figure 31 displays the compression rate over generations of a self-consuming training loop. A higher compression rate indicates a less diverse sample. Similar to our results in the logic expression domain we observe the highest increase in compression rate and therefore decrease in diversity for the full synthetic data cycle. The incremental data cycle also has a stark increase in compression rate followed by the balanced data cycle. The expanding data cycle also shows an increase in compression rate, however, not as severe as for the other data cycles. This indicates that fresh data can slow down the effects of the self-consuming training as observed for the logic expression dataset.

Figure 32 displays the vocabulary size measured as unique words within a sample. Similar to the previous results we observe a decline in vocabulary size for all data cycles. While the full synthetic data cycle fully collapses in vocabulary size, closely followed by the balanced, and incremental data cycle, the expanding data cycle stabilizes around 500 words. However, this is still a drop in vocabulary size by 95%. Once again, this confirms a strong decline in diversity for all data cycles.

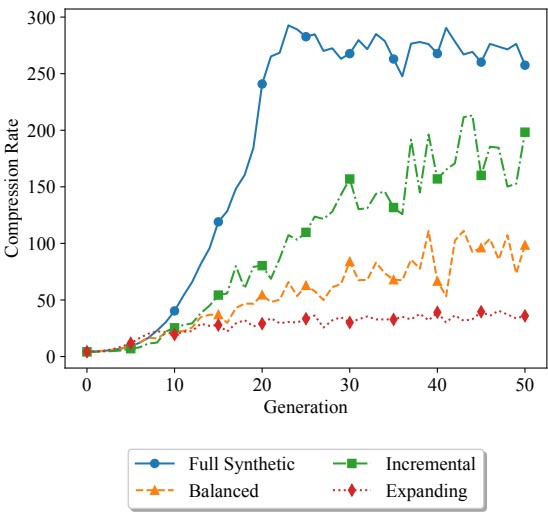
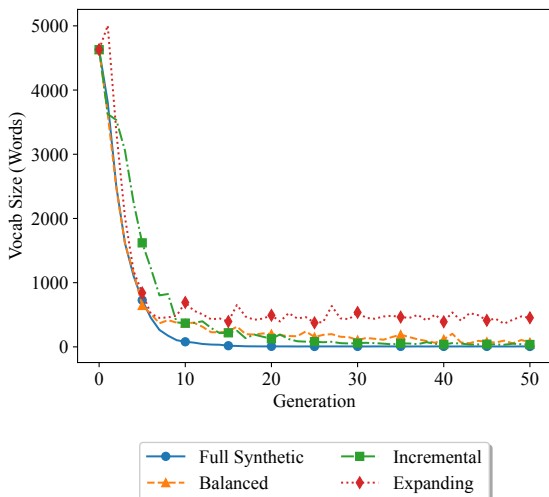

Figure 31: Compression rate over generations of a self-consuming training loop for different data cycles for textual data. A higher compression rate indicates less diversity. The sample size is truncated to equal length for better comparability. Line markers added every 5 generations for better display.

Figure 32: Vocabulary size over generations of a self-consuming training loop for different data cycles for textual data. The vocabulary size is defined as the number of unique words in a sample. The sample size is truncated to equal length for better comparability. Line markers added every 5 generations for better display.

## E  Example Outputs

### E.1  Examples of Logic Expressions

Table 2 provides examples of logic expressions that evaluate to `True` or `False`, as well as syntactically incorrect logic expressions (`Error`). In the erroneous example the logic expression is syntactically incorrect because of a missing parentheses. Additionally, all generated logic expressions from our experiments are provided in the supplementary material.[6]

Table 2: Examples of logic expressions that evaluate to `True`, `False`, or cannot be evaluated (`Error`).

| TYPE | EXPRESSION |
|------|------------|
| True | ( not ( ( ( True and True ) and ( not True ) ) or ( ( False or False ) and ( not True ) ) ) ) <eos> |
| False | ( ( ( ( True and False ) and ( True or False ) ) or ( not ( False or True ) ) ) and ( ( ( True and True ) and ( True or False ) ) or ( ( True or True ) and ( True or True ) ) ) ) <eos> |
| Error | ( ( ( False or True ) and ( not True ) or ( ( not True ) or ( not False ) ) ) <eos> |

---

[6]https://figshare.com/s/b09d5bdb3b216998d330 - Anonymized for review

### E.2   Examples of Natural Language

Figure 33-36 provide example outputs of models trained in a self-consuming training loop on the natural language dataset for all data cycles. Displayed is an example of the initial data and examples for generations 1, 10, and 50. The examples are truncated for better display. All generated text samples for each generation and data cycle are also provided in the supplementary material.[6]

We manually inspect the text samples for the first few generations and notice a few hallucinated words that are not present in the original corpus, indicating a drop in correctness. However, overall the generated text is subjectively mostly correct and Shakespeare like with a diverse set of samples.

By generation 10 the diversity is already declining for all data cycles. This is most severe for the full synthetic data cycle which produces nearly the same output every time. The incremental data cycle is also very repetitive text with repeating words in half of the time. In the other half (not shown in the example), the output is just a repetition of a role without text (e.g. `GLOUCESTER:\n\n GLOUCESTER:\n\n GLOUCESTER:\n\n`). The other two data cycles (balanced and expanding) show a strong repetition of words.

By generation 50 the full synthetic data cycle is fully converged only producing the same outputs. The incremental data cycle also shows heavy repetition and by generation 50 only very few roles are sampled repeatedly without text besides a very few exceptions. The other two data cycles continue the trend of producing repetitive words and $n$-grams and by generation 50 only small diverse text samples are generated before the same word or $n$-gram is produced repeatedly.

Overall, while one could argue that the text is more *correct* (fewer hallucinated words) at the end of a self-consuming training loop, the stark decrease in diversity and the generation of meaningless text makes the language model unusable.

**Full Synthetic Data Cycle**

**Initial:**
First Citizen:
Before we proceed any further, hear me speak.

All:
Speak, speak.

First Citizen:
You are all resolved rather to die than to famish?

- - - - - - - - - - - - - - - - - - - - - - - - - - - - - - - - - - - - - - - - - - - - - - - - - - - - - - - -

**Generation 1:**
YORK:
I think you where you have seen to be envious death.

LORD STANLEY:
No more, my lord: let me speak with men and my soul
Your proof interior marriage, with it end.

WARWICK:
Fair lords, and Warwick, where she lives you lies?

- - - - - - - - - - - - - - - - - - - - - - - - - - - - - - - - - - - - - - - - - - - - - - - - - - - - - - - -

**Generation 10:**
WARWICK:
What with his wisdom?

WARWICK:
Why, what with his soul?

WARWICK:
What with his soul?

- - - - - - - - - - - - - - - - - - - - - - - - - - - - - - - - - - - - - - - - - - - - - - - - - - - - - - - -

**Generation 50:**
WARWICK:
What with his wisdom?

WARWICK:
What with his wisdom?

WARWICK:
What with his wisdom?

Figure 33: Example model outputs of the full synthetic data cycle for the natural language experiments over generations of a self-consuming training loop.

**Balanced Data Cycle**

**Initial:**
First Citizen:
Before we proceed any further, hear me speak.

All:
Speak, speak.

First Citizen:
You are all resolved rather to die than to famish?

- - - - - - - - - - - - - - - - - - - - - - - - - - - - - - - - - - - - - - - - - - - - - - - - - - - - -

**Generation 1:**
ANGELO:
Anon I would have no more for your better.

ANGELO:
I will be so not better, whom you flow.

ISABELLA:
I would he that would see your behalf.

- - - - - - - - - - - - - - - - - - - - - - - - - - - - - - - - - - - - - - - - - - - - - - - - - - - - -

**Generation 10:**
CORIOLANUS:
You will not speak to the people of the prince of the people,
Be so she is a man in the prince of the prince,
And the traitor of the prince of the prince,
And the prince of the prince of the prince,
And the prince of the prince of the prince of the prince of the prince,
And the seat of the prince of the state of the prince of the prince,
And so she shall be so she is not so shall not speak to the prince of the prince of the prince,
And the seat of the seat of the state of the prince of the state of the state,

- - - - - - - - - - - - - - - - - - - - - - - - - - - - - - - - - - - - - - - - - - - - - - - - - - - - -

**Generation 50:**
Here's noble by so dead.

DUKE VINCENTIO:
Here's a man in the prince of the prince, that the prince of the prince of the prince,
And the prince of the prince of the prince of the prince,
And the prince of the prince of the prince of the prince,
And the prince of the seat of the prince of

And the state of the state of the state of the state, ... *(continues)*

Figure 34: Example model outputs of the balanced data cycle for the natural language experiments over generations of a self-consuming training loop.

**Incremental Data Cycle**

**Initial:**
First Citizen:
Before we proceed any further, hear me speak.

All:
Speak, speak.

First Citizen:
You are all resolved rather to die than to famish?

------------------------------------------------------------

**Generation 1:**
TYBALT:
I'll stand my lord.

Marshal:
Tell her her and be solemned and soldiers,
For her wrong so understrange for Lancaster.

DUKE VINCENTIO:
Go you, good good lady.

------------------------------------------------------------

**Generation 10:**
LADY CAPULET:
I will not say the world,
And thou hast not said to be so for the world,
And with the world.

LADY CAPULET:
I have said to shame the world,
And thou hast not said to show the world,
And thou shalt not say the world in the world,
That I have not said to shame the world.

------------------------------------------------------------

**Generation 50:**
DUKE OF YORK:

DUKE OF YORK:

DUKE OF YORK:

Figure 35: Example model outputs of the incremental data cycle for the natural language experiments over generations of a self-consuming training loop.

---

**Expanding Data Cycle**

**Initial:**
First Citizen:
Before we proceed any further, hear me speak.

All:
Speak, speak.

First Citizen:
You are all resolved rather to die than to famish?

- - - - - - - - - - - - - - - - - - - - - - - - - - - - - - - - - - - - - - - - - - - - - -

**Generation 1:**
SICINIUS:
The enemy noble are and true,
And like it to life he putinion.

VOLUMNIA:
So, let not for us.

CORIOLANUS:
No, I wise the take tower?

- - - - - - - - - - - - - - - - - - - - - - - - - - - - - - - - - - - - - - - - - - - - - -

**Generation 10:**
LADY ANNE:
Would good we have my most for my soul, but your shouls
The can is sould distrother particularish,
And the such that the take tribunes,
Where the name and the vail the people and to still grave the truest roper in the people,
Which the port the people that who the hath the shall should to the people the people the people
The people of the people and to the people tribuness of the people tribuness, the people,

- - - - - - - - - - - - - - - - - - - - - - - - - - - - - - - - - - - - - - - - - - - - - -

**Generation 50:**
LADY ANNE:
The gentleman is good more gone?

GLOUCESTER:
You have been my lord, I will be possess to the gods gods!

GLOUCESTER:
What the grace the people the people the people the people the people the people the people the people the people the people the people the people the people the people the people *. . . (continues)*

Figure 36: Example model outputs of the expanding data cycle for the natural language experiments over generations of a self-consuming training loop.

