# OpenReview forum: "Large Language Models Suffer From Their Own Output: An Analysis of the Self-Consuming Training Loop"
_TMLR — Rejected by TMLR_

### Review · Reviewer_6D2Y · 2025-08-23

**Summary Of Contributions:**

The paper studies the self-consuming setting of LLM's training, where the model is trained on both human written data and LLM generated data. It examines the quality of generation in terms of diversity and correctness on a synthetic logic expression dataset and Shakespeare dataset. The experiment results show that the correctness is improved over the training, yet the diversity of generation diminishes in terms of  Levenshtein distance and N-gram diversity score.

**Audience:**

Yes

**Audience Explanation:**

The setting is meaningful and should be a interest of broader research community in natural language processing.

**Broader Impact Concerns:**

Not applied.

**Claims And Evidence:**

Yes

**Claims Explanation:**

The claim is validated in 10 million size language model under two simple datasets. However, generalizing the claim into more practical setting requires more comprehensive experiments on practical setting, including various model scales, data scales, and more fine-grained metrics.

**Requested Changes:**

The work's experiment setting deviates from the practical LLM training by a large margin, which makes the result not very convincing to me. The following change are suggested.

* **Scale up model size.** The study's experiments are conducted on models of 10 million parameters, which is significantly smaller than the large-scale LLMs where the self-consuming loop is a primary concern. The dynamics of billion-parameter models can differ substantially, which may limit the direct applicability of these findings. To strengthen the paper's claims, it would be beneficial to demonstrate that the initial trend of diversity decay also occurs on a larger, publicly available model, even if for fewer generations.

* **Experiment on different data domains.** The paper's findings are derived from a synthetic logic expression dataset and the stylistically narrow TinyShakespeare corpus. While the logic task offers clear evaluation, its simplicity may not fully represent the complexities of natural language. It is recommended that the authors validate their results on a more diverse and complex text corpus to show that the observed effects are not an artifact of the simplified experimental domains.

* **Include more comprehensive metrics.** The analysis relies on metrics that primarily measure surface-level syntactic variety, such as Levenshtein distance and n-gram diversity. These may not fully capture the loss of deeper semantic or conceptual diversity. The study would be more robust if it incorporated embedding-based metrics to provide a more complete picture of semantic collapse alongside the observed syntactic degradation.

---

> ### Author Response · Authors · 2025-10-09
> **Response to reviewer 6D2Y**
>
> Dear reviewer 6D2Y,
>
> Thank you for taking the time to review our work and for providing valuable feedback and suggestions.
>
> We will address your concerns below:
>
> - We are aware of the limitations regarding model size and dataset size in comparison to modern, state-of-the-art LLM training and consequently expanded the discussion of this limitation in the manuscript. Furthermore, to address these concerns, we conducted studies with various model sizes up to 49.2M and different data set sizes in the Appendix. For the full synthetic data cycle, we present results for four different models ranging from 3.2M up to 49.2M (appendix D.4) and for three different numbers (appendix D.5) of logic expressions (20,000, 30,000, and 40,000). Across **all** experiments we found a decline in diversity regardless of model size and dataset size.
>
>   We are aware that this will perhaps not fully alleviate the concerns of the reviewer, however, we do not see any chance to perform an analysis of the self-consuming loop for current state-of-the-art models with some billion parameters as well as realistic training data sets. Even training a state-of-the-art model only once will already cost millions of Dollars which prohibits scaling up the analysis to dozens or hundreds of training cycles for billion parameter models. Even more, training such huge models is not at all responsible in terms of the expected energy consumption as well as CO2 emissions.
>
>   At the moment, we provide experimental results for models up to 49.2M parameters. For **all** studied models, we observe a loss of diversity. Thus, we are convinced that we can interpolate the results and also larger models will be affected. We as a machine learning community must provide answers for a world where already a lot of content which is used for the training of current state-of-the-art models is not fresh data but produced by models. The work at hand studies the self consuming loop and finds that the resulting loss of diversity is not only a problem for the generation of images but also for the generation of text. We believe these results are relevant for the community even if generating results for state-of-the-art models is far beyond our capabilities.
>
>   In summary, we do not think that it is feasible nor responsible in terms of computational resources to conduct a large-scale experiment over multiple generations with 1B+ Models and corresponding datasets. Therefore, in line with the comments of the reviewers as well as TMLR’s policies for acceptance criteria, we carefully adjusted the claims throughout the manuscript (particularly in abstract, introduction, limitations, and conclusion) to reflect that our results are bound to the experiments at hand and that our hypotheses about larger models are only extrapolations from these results.
>
>   As a side comment, we want to point out that it might be possible to increase the size of the models (as well as the size of the data sets) to for example 100M parameter models as well as to larger data sets. But of course, this would come with a ridiculously high effort for training such large models, prohibiting it. Nevertheless, even when providing results for 100M models, the experiments would still be far away from current state-of-the-art models with many Billion parameters making the endeavor of substantially increasing the size of the model not reasonable nor responsible.
>
> - Regarding your suggestions about additional metrics: We were happy to follow this suggestion and consequently extended the main findings of the paper with a set of additional metrics in line with your (and reviewer hiiK’s) suggestions. We provide embedding based cosine distances for both logic expressions (Appendix D.8) and textual data (Sect. 4.3) as well as several other metrics for logic expressions in Appendix D.8. Across those metrics we observe a similar decline in diversity as before. We believe this provides a more comprehensive view on the decline of diversity in our paper.
>
> We incorporated these comments and additional results as well as the suggestions of the other reviewers into the manuscript and marked the corresponding changes in blue.
>
> We hope this addresses your concerns.
>
> Sincerely
> The author(s)

---

### Review · Reviewer_hiiK · 2025-08-28

**Summary Of Contributions:**

The paper investigates self-consuming training loops for LLMs trained from scratch across generations (not iterative fine-tuning). Using a synthetic logic-expression corpus with exact correctness checks and four data-cycle regimes—full-synthetic, balanced, incremental, expanding—the authors show: 1- measured “correctness” can rise across generations, while 2) output diversity declines, sometimes collapsing; 3) adding fresh data slows the decline; 4) similar trends appear on tiny-Shakespeare text. The setup is clearly specified with algorithms, model/training details, and extensive appendices

**Audience:**

Yes

**Audience Explanation:**

Yes. The question is timely, the setup is clean, and findings are decision-relevant for data pipeline design.

**Broader Impact Concerns:**

The work spotlights a real risk for the LLM ecosystem: synthetic feedback can degrade future generations, potentially harming quality and trust. Consider adding a short mitigations section (data provenance tracking, filtering/weighting synthetic data, decontamination checks, periodic real-data refresh, deduplication policies).

**Claims And Evidence:**

Yes

**Claims Explanation:**

Mostly yes, with caveats. Evidence solidly supports diversity decline under self-consumption and that more fresh data slows it. The statements that fresh data does not stop the decline and that collapse is eventual are plausible but extend beyond T=50; they should be framed explicitly as extrapolations.

**Requested Changes:**

Critical
1.	Re-evaluate the “correctness” story.
  -Add runs where D₀ is balanced True/False (or otherwise distributionally specified) and report correctness against that target.
- Complement token edit distance with structure-aware metrics (e.g., AST/operator distributions, length distributions).
- Note: Appendix D.2’s 10-generation True/False/mixed initializations are a good start; extending these to T=50 and foregrounding mixed D₀ in the main text would clarify conclusions.
2.	Sampling ablation.
- Repeat at least one full data-cycle experiment with varied temperature and top-p for generating S_t to measure sensitivity of collapse timing/magnitude.
3.	Deduplication ablation.
- State explicitly whether S_t/D_t are deduplicated each generation, and add a with- vs. without-dedup comparison to isolate duplication effects from model-driven collapse.

Nice-to-have

4.	Longer horizon or theory to support “eventual collapse,” or soften to an empirically bounded claim.
5.	Broader diversity views: Self-BLEU, token entropy over time, AST/tree-shape and operator histograms, and KL to a held-out real dataset. For text, include word-level n-grams in addition to character-level.
6.	Reproducibility upgrades: release code, configs, seeds, and precise GPU details; consider releasing per-generation corpora and sampling metadata.

---

> ### Author Response · Authors · 2025-10-09
> **Response to reviewer hiiK**
>
> Dear reviewer hiiK,
>
> Thank you for your valuable feedback and attention to detail. Below we want to address your comments point by point.
>
> - According to your suggestions we extended our experiments with additional results for runs with equal proportions of True and False expressions. We find similar results to the previous experiments in terms of decline in diversity. Additionally, we observe that the initial dataset can have an effect on the decline in diversity. In terms of correctness we find that the self-consuming training loop shifts the distribution away from the original proportions. We incorporated these results into the manuscript (Sect. 4.1 and 4.2) and adjusted our claims accordingly throughout the manuscript.
>
> - Additionally, we provide complementary metrics according to your suggestions in Appendix D.8. We investigate embedding based cosine distance (as suggested by reviewer 6D2Y), token entropy, AST depth and number of AST nodes, and the distribution of tokens per expression over generations. Across those metrics we observe similar findings as with the Levenshtein distance.
>
> - Furthermore, we conducted a set of ablation studies for sampling parameters in Appendix D.6. We find that lower temperatures values lead to a quicker decline in diversity and higher values lead to a slower decline. If the temperature value is large enough the diversity actually increases, however, this is due to an increasing amount of false expressions due to errors that accumulate over the course of the self-consuming training loop. For top-p we find that lower values of top-p lead to a quicker decline in diversity, which is expected as we narrow the distribution of possible tokens.
>
> - Regarding your question about deduplication: We did not deduplicate the training data in our experiments. To address your concerns about the interactions of deduplication and the self-consuming training loop we conducted an ablation study with and without deduplication in Appendix D.2. We find that deduplication indeed can help with diversity preservation but does not stop the decline sufficiently.
>
> - We adjusted our claims in the abstract, introduction, limitations, and conclusion to make sure that the scope of our findings is restricted to our empirical results and that our hypotheses about the behavior of larger models are extrapolations of our results. We hope this addresses your comment as well as the concerns of reviewer 6D2Y and U4fv.
>
> - As mentioned in point 1.1 we added additional metrics to the Appendix. We also want to note that the n-gram diversity in Sect. 4.3 is on word level. We added this to the text to avoid further confusion.
>
> - We updated the GPU details for the prior and new experiments. Additionally, we made sure to provide further details on sampling and deduplication in Sect. 3.5. Regarding the corporas, they should be already present in the supplementary material. We updated this supplementary material with the new experiments as well. Lastly, we commit to releasing the code with publication of the manuscript.
>
> - We extended our discussion section with potential mitigation steps as suggested by you.
>
> We incorporated all changes into the new manuscript and marked those changes in blue color.
>
> We hope this addresses your questions and requests.
>
> Sincerely
> The author(s)

---

### Review · Reviewer_U4fv · 2025-09-26

**Summary Of Contributions:**

This work systematically analyzes the effect of a self-consuming training loop for LLMs, i.e., using LLM-generated content for training. Instead of an iterative fine-tuning approach, this study analyzes from-scratch training across generations to better simulate pretraining workflows in practice. Main experiments include training on (1) curated, verifiable logic expression dataset and (2) tiny Shakespeare dataset for testing effects on natural language. Experiments on both datasets show that such training decreases diversity and mixing with fresh real data slows but does not stop the decline.

**Audience:**

Yes

**Audience Explanation:**

1. The paper investigates a timely and important issue: the rise of LLM-generated content and risks/effects of self-consuming training cycles.
2. While similar studies have been conducted in generative models, the effects of self-consuming training is unclear in LLMs.

**Broader Impact Concerns:**

No broader impact concerns.

**Claims And Evidence:**

Yes

**Claims Explanation:**

**Strengths**

1. Clear framing with controlled experiments, positioned distinctly from prior works on investigation through iterative finetuning.
2. Provides analytical insights through varying perspectives, degree of model-generated data in training dataset, and diversity measures.

**Weakness**

1. The experiments are conducted on relatively small models (10.6M-nanoGPT; and 3.2M, 25.2M, 49.2M in Appendix), which remain far below realistically employed LLM model scales in both the training stage and content generation.
2. The use of a single base model for data generation and training doesn't fully capture real-world scenarios. For LLM training in practice, it is probably the case that multiple LLMs (with different architectures and learned distributions) generate contents, that are then used for training for the next generation of varying LLMs.

**Requested Changes:**

1. It would strengthen the work significantly if additional experiments could be conducted on self-consuming training mixed using a few different base models, as this further aligns with real-world scenarios.
2. I am further curious about how the initial dataset influences the findings. For instance, are there correlations between specific characteristics of the initial dataset and the extent of degradation observed during self-consuming training? I believe these analyses could provide insights for future research on improving the design and curation of training datasets.

---

> ### Author Response · Authors · 2025-10-09
> **Response to reviewer U4fv**
>
> Dear reviewer U4fv,
>
> Thank you for your valuable feedback and recognition of the importance of our work.
>
> In line with your comments we made several changes to the manuscript. These changes (along with the changes requested by the other reviewers) are marked in blue in the new manuscript.
>
> Below we want to address your points directly:
>
> - We are aware of the limitations of our study and mention them in the corresponding limitation section. However, we agree with the reviewers that claims about larger models are extrapolations of our results. While already mentioned in the original manuscript we made sure to carefully adjust our claims throughout the manuscript to reflect this.
> - We agree that in practice a range of LLMs are trained and produce data in parallel. To address this we conducted an additional controlled experiment in Appendix D.7. Starting from the same dataset we train three models of different sizes in parallel. We then sample one third of the new training dataset from each model and train the models again using that dataset. We find that all three models decline in diversity but the rate of this decline is slightly slower than for the individual models indicating that a diverse set of models can slow down the observed effects.
> - We conducted new experiments using an initial dataset of equal proportions of True and False expressions (as before in the Appendix in smaller scale) according to the suggestions of reviewer hiiK. We find that the initial diversity of data can indeed have an influence on the decline in diversity. We updated our claims and findings accordingly throughout the manuscript. While we do not have exact correlations for those influences we believe this is a promising direction for future work.
>
> Thank you again for your interesting suggestions and we hope this answers your questions.
>
> Sincerely
> The author(s)

---

### Author Response · Authors · 2025-10-09
**Response to reviews**

Dear reviewers and AE,

We would like to thank everyone involved in the review process for your effort and valuable feedback on our manuscript.

We are happy that all reviewers found our contribution meaningful and of interest to the TMLR community and our claims to be supported by clear and convincing evidence.

All reviewers gave some valuable feedback and in line with the suggestions of the reviewers we **updated our manuscript and supplementary material including new experimental data and carefully adjusting our claims.**

To make it easier for the reviewers all corresponding changes in the new manuscript are marked in blue color. We also provide a response to each reviewer below.

**We believe that this revision substantially strengthened our paper and that we have addressed the reviewers' concerns.**

Thank you for your continued engagement and do not hesitate to ask further questions.

Sincerely
The author(s)

---

### Decision · Action_Editor_BNiN · 2025-11-27

**Recommendation:** Reject

**Audience:**

Yes

**Audience Explanation:**

The paper addresses a timely and significant topic: when LLMs are trained on their own output, there might be a risk of "model collapse". The topic of this paper will be interesting to many readers.

**Claims And Evidence:**

No

**Claims Explanation:**

While the paper studies a very interesting topic and presents many interesting results, the reviewers generally find that the current state of the paper doesn't sufficiently support the major claim (i.e., whether LLMs suffer from mode collapse due to self-consuming training loops). There are two major concerns raised by the reviewers (and not successfully addressed in the authors' response):

1. The model scales used in the experiments are too small.
One critical issue, raised by both Reviewer U4fv and Reviewer 6D2Y, is the significant disconnect between the paper's title/claims (LLMs) and the experimental setup: The experiments rely on "NanoGPT" style models, initially around 10M parameters, and scaled up to only ~49M parameters in the rebuttal. The datasets used are synthetic logic expressions and "Tiny Shakespeare."
Reviewer U4fv and Reviewer 6D2Y argue that these models are too small to represent the dynamics of actual LLMs (usually at least 1B+ parameters). LLMs often exhibit emergent behaviors that small models do not; therefore, concluding that LLMs will inevitably collapse based on <50M parameter models is an unsupported extrapolation.
The authors argued that training billion-parameter models from scratch for 50 generations is computationally infeasible for academic labs. While this is a valid constraint, the reviewers felt that the gap (from 49M to 1B+) was too large to sustain the paper's core claims about the behavior of state-of-the-art systems.

2. The nature of the datasets is oversimplified.
Reviewer 6D2Y highlighted that the "Over-simplified setup" (synthetic logic and stylistically narrow Shakespeare text) deviates heavily from the pre-training data used for actual LLMs. While the logic dataset offers the benefit of objective verification (a strength noted by the authors), the reviewers were not convinced that findings on such restricted domains generalize to the complex, open-domain nature of modern LLM training.

Suggestions for revisions: The authors should scale up their models significantly (at least to the ~1B range). Given the authors' access to A100 GPUs, it should be affordable to at least fine-tune ~1B LLMs (it's also fine to observe the model behaviors through fine-tuning instead of pre-training from scratch) with mid-scale datasets (so that the costs are affordable). Furthermore, incorporating more open-domain and real-world text datasets will strengthen the claims.

**Resubmission Of Major Revision:**

The authors may consider submitting a major revision at a later time.